# varVAMP: degenerate primer design for tiled full genome sequencing and qPCR

Jonas Fuchs [1] ✉, Johanna Kleine[1], Mathias Schemmerer [2], Julian Kreibich [3], Wolfgang Maier[4], Namuun Battur[5], Thomas Krannich [5], Somayyeh Sedaghatjoo[5], Lena Jaki [1], Anastasija Maks[1], Christina Boehm[2], Carina Wilhelm [2], Jessica Schulze[6], Christin Mache[6], Elischa Berger[4], Jessica Panajotov [7], Lisa Arnold[8], Björn Grüning [4], Markus Bauswein[8], Sindy Böttcher[3], Reimar Johne[7], Jürgen Wenzel [2], Martin Hölzer[5] & Marcus Panning [1] ✉

Time- and cost-saving surveillance of viral pathogens is achieved by tiled sequencing in which a viral genome is amplified in overlapping PCR amplicons and qPCR. However, designing pan-specific primers for viral pathogens with high genomic variability represents a significant challenge. Here, we present a bioinformatics command-line tool, called varVAMP (variable virus amplicons), which addresses this issue. It relies on multiple sequence alignments of highly variable virus sequences and enables degenerate primer design for qPCR or tiled amplicon whole genome sequencing. We demonstrate the utility of var-VAMP by designing and evaluating novel pan-specific primer schemes suitable for sequencing the genomes of SARS-CoV-2, Hepatitis E virus, rat Hepatitis E virus, Hepatitis A virus, Borna-disease-virus-1, and Poliovirus using clinical samples. Importantly, we also designed primers on the same input data using the software packages PrimalScheme and Olivar and showed that varVAMP minimizes primer mismatches most efficiently. Finally, we established highly sensitive and specific Poliovirus qPCR assays that could potentially simplify current Poliovirus surveillance. varVAMP is open-source and available through PyPI, UseGalaxy, Bioconda, and https://github.com/jonas-fuchs/varVAMP.

In recent years, next-generation full-genome sequencing of viruses has become an irreplaceable method for tracking the evolution of viral pathogens, studying outbreaks in the human population and animal kingdom, and identifying novel zoonotic threats[1–3]. While metagenomic analyses enable the broad analysis of viromes and potentially identify novel pathogens[4], high genome coverage is needed to sufficiently analyze the genomic makeup of a viral population[5,6]. This can be achieved by prior virus cultivation or increased sequencing depth, which have drawbacks. Virus cultivation is not always successful and can lead to cell-culture adaptations[7]. Moreover, increased sequencing depth is costly and might still result in poor genome coverage[8]. Targeted sequencing approaches via PCR-

[1]Institute of Virology, Freiburg University Medical Center, Faculty of Medicine, University of Freiburg, Freiburg, Germany. [2]Institute of Clinical Microbiology and Hygiene, National Consultant Laboratory for HAV and HEV, University Medical Center Regensburg, Regensburg, Germany. [3]National Reference Center for Poliomyelitis and Enteroviruses, Robert Koch Institute, Berlin, Germany. [4]Bioinformatics Group, Department of Computer Science, Albert-Ludwigs-University Freiburg, Freiburg, Germany. [5]Genome Competence Center (MF1), Robert Koch Institute, Berlin, Germany. [6]Unit 17 "Influenza and Other Respiratory Viruses", Robert Koch-Institute, Berlin, Germany. [7]Department of Biological Safety, German Federal Institute for Risk Assessment (BfR), Berlin, Germany. [8]Institute of Clinical Microbiology and Hygiene, University Hospital Regensburg, Regensburg, Germany. ✉e-mail: jonas.fuchs@uniklinik-freiburg.de; marcus.panning@uniklinik-freiburg.de

tiling or DNA hybridization allow highly specific sequencing on smaller machines without prior pathogen cultivation[9,10]. In particular, PCR-tiling, in which the viral genome is amplified in overlapping fragments, has gained popularity because of its cost-effectiveness, low input requirement, and simple library preparation. The most prominent viral amplicon schemes were developed for SARS-CoV-2 in early 2020 and have allowed the sequencing of millions of viral genomes during the pandemic[11,12]. However, such amplicon schemes often need to be updated to reflect evolutionary changes or they have not been developed at all for many viral pathogens. Therefore, quantitative real-time PCR (qPCR) remains the diagnostic gold standard for analyzing patient samples for the presence of a viral pathogen[13].

In an optimal setting, tiled-sequencing and qPCR primer designs for viral pathogens should be pan-specific. This can be challenging for viruses with a high genomic variability and common insertion and deletion (INDELs) sites. Thus, primers must be designed in conserved regions with minimal genomic variation and should not span INDELs. As potential primer target regions might still display sequence variation, degenerate nucleotides can be introduced into primer sequences to further broaden their binding capacity. Optimal pan-specific primers must target highly conserved regions while keeping degeneracy minimal. This problem, termed maximum coverage degenerate primer design (MC-DGD), is a trade-off between specificity and sensitivity[14]. Primer-specific parameters complicate MC-DGD as not all potential regions are also potential primer binding sites[15]. Notably, qPCR designs have even greater constraints due to additional hydrolysis probe-specific parameters and a low Gibbs free energy change ($\Delta G$) in the target region[16].

Various commercial and open-source primer design applications are available and often utilize primer3 at their core to calculate various primer parameters[17]. However, many of these tools were developed for a particular primer design problem, and each only addresses some of the previously mentioned problems[18]. PrimalScheme is the gold standard for designing tiled primer schemes for viral full genome sequencing[10]. However, PrimalScheme was not developed to handle highly divergent alignments such as Hepatitis E virus (HEV) or Hepatitis A virus (HAV)[19]. The recently published software package Olivar addresses variant-aware primer design for tiled sequencing by minimizing a primer's risk score which incorporates information about sequence variations at a given primer position[20]. However, both Olivar and PrimalScheme do not introduce degenerate nucleotides into primer sequences or design multiple discrete primers to compensate for mismatches, limiting or even abolishing the binding affinity if variants within a primer sequence are unavoidable owing to overall high sequence variability[21]. Software packages such as easyPAC or DegePrime[22,23] have elegantly addressed primer design, but they are not suited for the automatic design of tiled or qPCR schemes. For qPCR primer design, there are only a few open-source projects like QuantPrime[24], but most software is not open-access and is available through commercial companies. However, none of these applications address pan-specific primer design, and not all methods calculate $\Delta G$, resulting in time-intensive manual primer and amplicon evaluation.

Here, we present the command-line tool varVAMP (variable virus amplicons) which enables degenerate primer design for single amplicons, tiled amplicon schemes and qPCR and is tailored to viral genomics. We show varVAMP's utility by designing and testing pan-specific tiled and qPCR primer sets for SARS-CoV-2, HEV (*Paslahepevirus balayani*), ratHEV (*Rocahepevirus ratti*), HAV (*Hepatovirus A*), Borna-disease-virus 1 (BoDV-1, *Orthobornavirus bornaense*), and Poliovirus (*Enterovirus C*, PV) 1-3, which represent different levels of sequence variability. Moreover, we analogously design primer schemes for all viruses via both Olivar and PrimalScheme and show that varVAMP minimizes primer mismatches most efficiently.

## Results

### Software and output

The command-line tool varVAMP was written in python3 and requires only a precomputed MSA as input. Notably, varVAMP is cross-platform (Windows 11, MacOS and Linux) with Python 3.9 or higher being the single requirement prior to installation. varVAMP can design primers for single amplicons, tiled amplicon schemes and qPCR. The pipeline consists of multiple steps that are common to all different modes (alignment preprocessing, consensus generation and primer evaluation), mode-specific or optional (automatic parameter search and BLAST evaluation) (Fig. 1a). At its core, varVAMP wraps Primer3[17] and uses a k-mer-based approach to find all potential primers in a consensus sequence calculated from the input MSA. varVAMP addresses the MC-DGD problem by first calculating two consensus sequences that consist either of the majority nucleotides at each position or integrate degenerate nucleotides. The latter is used to find potential primer regions that are regions with a user-defined maximum number of degenerate nucleotides within the minimal primer length. Afterwards, k-mers of the majority consensus sequence that lie within these potential primer regions are tested for all relevant primer parameters. varVAMP evaluates these primers via a penalty system that incorporates information about primer parameters, 3' mismatches, and degeneracy. In its tiled sequencing mode, varVAMP finds overlapping amplicons spanning the alignment while minimizing primer penalties using Dijkstra's algorithm to find the shortest paths between nodes in a weighted graph[25]. For qPCRs, varVAMP independently evaluates probe and primer parameters and tests the $\Delta G$ of potential qPCR amplicons. The final primers are then deduced from the consensus sequence incorporating degenerate nucleotides. varVAMP produces multiple outputs in standardized formats and a plot displaying the alignment's normalized Shannon's entropy, all potential target regions, all primers that passed the initial filtering steps and the final amplicon design with low penalty primers (Fig. 1b).

### Design and evaluation of HEV pan-specific primers

HEV of the genus *Paslahepevirus* is the most common cause of acute viral hepatitis worldwide and is phylogenetically separated into four distinct genotypes (genotypes 1–4). In risk groups such as immunocompromised patients, zoonotic HEV genotype 3 (HEV-3) can cause acute or chronic hepatitis[26]. HEV-3 has a high prevalence in industrialized countries and is further classified into subgenotypes with varying prevalence rates depending on the geographic region[27]. Most genome sequences show exceptional variability[28] and must be generated from the initial patient material, as virus isolation requires optimized cell culture systems[29]. To provide a simple sequencing procedure from patient material and to test the real-world applicability of varVAMP, we set out to design primers for HEV-3 tiled sequencing as a proof-of-principle. We initially downloaded all available full-genome HEV sequences from the NCBI GenBank and classified the (sub-) genotypes using fasta36 as previously described (Fig. 2a)[2]. Our aim was to design primers that would be specific for multiple HEV-3 sub-genotypes. Therefore, sequences were clustered based on their similarity via vsearch[30] and the clustering result was evaluated by constructing a maximum-likelihood phylogenetic tree with IQ-TREE 2[31]. Clustering resulted in seven clusters with more than 6 sequences (Fig. 2b). Four large clusters belong to HEV-3 each comprising multiple sub-genotypes. We designed primers for cluster 2 (HEV-3 f, e) and cluster 4 (HEV-3 c, h1, m, i, uc, l) to reflect the most common European HEV-3 subgenotypes[2]. Therefore, the sequences of cluster 2 and 4 were separately aligned with MAFFT[32] and the alignments used as the input for varVAMP yielded an amplicon design of seven and six 1–1.5 kb amplicons, respectively (Fig. 2a and Table 1). Next, we evaluated these primer schemes on persistently HEV-3 f- (strain: 15-22016) and c- (strain: 14-16753) infected cell cultures via an one-step RT-PCR protocol. Agarose gel electrophoresis revealed consistent and strong

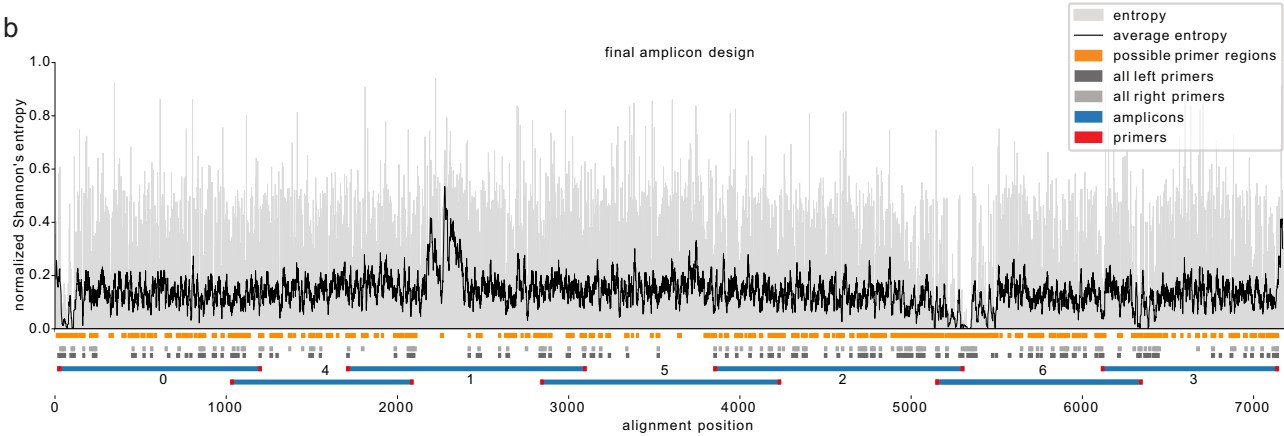

**Fig. 1 | Schematic overview of the varVAMP workflow and example output.**
**a** Overview of the varVAMP workflow. The white boxes represent steps of the pipeline that are common to all modes. The consecutive steps are connected by arrows, and the optional steps are indicated with a dotted border. Colored boxes mark unique steps for each varVAMP mode (blue–single, orange–tiled, green–qPCR). Steps that produce outputs end in schematic folder icons for the main output and the additional data subfolder. (n number, nt nucleotide). Created in BioRender. Fuchs (2025) https://BioRender.com/2l4hzpe. **b** Example overview plot that is produced when running varVAMP. This plot was generated with varVAMP tiled mode on the example MSA of HEV-3 sequences provided as example data within the varVAMP github repository. The normalized Shannon's entropy for each alignment position (gray) and its rolling average over 10 nucleotides (black curve) are shown. The orange boxes below the plot mark the start and stop MSA positions of potential primer regions (regions that have, in this case, a maximum of 4 ambiguous bases within the minimal primer length of 19) and the gray and light gray boxes mark all considered forward and reverse primers, respectively. The final scheme that was selected by the graph search for overlapping amplicons (blue) with low-penalty primers (red) is depicted at the bottom.

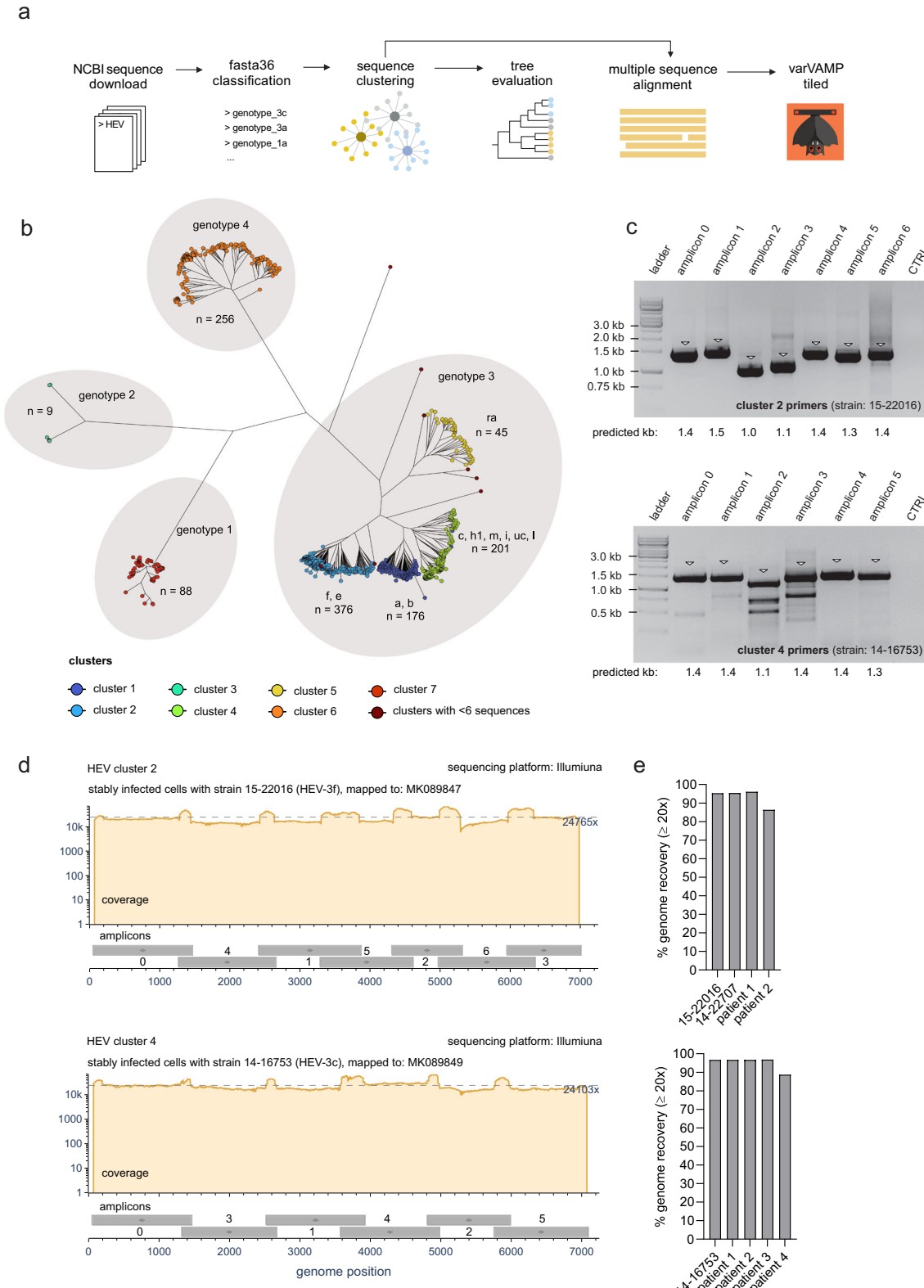

amplification for all amplicons with only a few nonspecific bands for amplicon 2 and 3 of cluster 4 (Fig. 2c). Next-generation Illumina sequencing of the pooled PCR products resulted in an even and high coverage for both samples (Fig. 2d). To further evaluate the primer schemes, we applied our protocol to a third HEV-3 e (strain: 14-22707) persistently infected cell culture for cluster 2 and to HEV-3 positive patient material for both clusters. In order to select the proper

amplicon scheme, we first subclassified HEV-3 positive blood samples. Next, we evaluated the cluster 2 and 4 primer schemes on the HEV-3 e (n = 2) and HEV-3 c (n = 4) samples, respectively. The next-generation sequencing results were comparable to prior results and allowed HEV-3 genome reconstruction (Fig. 2e and Supplementary Fig. 2). However, for patient 2 of cluster 2 and patient 4 of cluster 4, we observed a single amplicon dropout (Supplementary Fig. 2). Interestingly, both

**Fig. 2 | Primer design and tiled sequencing of HEV-3. a** Schematic overview of the data preparation steps preceding primer design. All full-length sequences of HEVs were downloaded from NCBI, sub-genotyped with fasta36 and clustered by similarity with vsearch. The clustering result was evaluated via phylogenetic tree construction. Clusters comprising multiple subgenotypes were aligned with MAFFT, and the MSA used as the input for varVAMP. Created in BioRender. Fuchs (2025) https://BioRender.com/csajk3b. **b** Phylogenetic tree of full-length HEV sequences constructed with IQ-TREE 2 (GTR + F + R10, 1000 bootstrap replicates). The vsearch clustering result for each sequence is displayed in colors and the HEV genotypes and subgenotypes are indicated at the respective branches (n number of sequences). **c** Agarose electrophoresis images of the individual PCR products for the cluster 2 (upper plot, representative plot out of 4 plots) and cluster 4 (lower plot, representative plot out of 5 plots) primer schemes tested with the supernatants of HEV-3 f or HEV-3 c stably infected PLC/PRF/5 cells, respectively. Triangles indicate

bands at the expected molecular weight of the PCR products (kb kilobases). **d** Coverage plots of the Illumina sequencing results of the in **c** amplified PCR products for cluster 2 (upper plot) and cluster 4 (lower plot) mapped to their respective NCBI reference sequences MK089847 and MK089849. Below each coverage plot, the genomic start and stop positions of each amplicon are displayed as gray boxes with their respective amplicon number. Dotted lines indicate mean coverages. Coverage plots were created with BAMdash (individual coverage plots are given in Supplementary Fig. 2). **e** Genome recovery of HEV-3 persistently infected cell cultures and sub-genotyped HEV-3 positive blood samples subjected to their respective tiled amplicon workflow for cluster 2 (upper plot) or cluster 4 (lower plot). Genome recovery was calculated as the percentage of reference nucleotides covered at least 20 fold. All PCRs were performed in the singleplex setting.

dropouts were caused by amplicons with a forward primer close to the HEV-3 hypervariable region that has an inherently higher sequence variability[28]. This suggested that rare or unique variations in this region might have restricted primer binding in isolated cases.

In summary, we used varVAMP to design two tiled primer schemes each specific for multiple HEV-3 sub-genotypes and used Illumina sequencing to recover near-to-complete viral genomes from both infected cell cultures and patient material.

## In silico evaluation of primer schemes for multiple viral pathogens and head-to-head benchmark to PrimalScheme and Olivar

We designed primer schemes for tiled full-genome sequencing for SARS-CoV-2, BoDV-1, HAV, PV and ratHEV, which display different degrees of sequence conservation over the whole genome with SARS-CoV-2 having the lowest (99% pairwise identity) and ratHEV the highest overall sequence variability (57% pairwise identity) (Table 1 and Fig. 3a). The initial data selection and preprocessing was highly dependent on the individual datasets and inspired by our experiences with HEV-3. For SARS-CoV-2, we did not directly align sequences from public databases but created representative consensus sequences of circulating lineages in Germany between October 2021 and September 2023 (920k samples) to represent the most prevalent variations for each lineage within the alignment. On the basis of the mean pairwise identities, we chose to tolerate one to five ambiguous bases within the primer sequences and optimized the identity threshold (Table 1). Except for BoDV-1 and SARS-CoV-2, we aimed for an amplicon size of more than 1000 bp so that amplicons could span regions with higher entropy in which potential primers are scarce (Fig. 3a).

Next, we evaluated the schemes in silico. First, we analyzed the degeneracy per primer, which is highly penalized by varVAMP to minimize the number of permutations. Two, four, and five tolerated ambiguous bases within a primer sequence can lead to maximum of 4, 256, and 1024 primer permutations, respectively. However, for the primer schemes with four and five tolerated ambiguous bases, the mean number of permutations was over 10-fold lower than theoretically possible, indicating preferential selection of primers with low degeneracy (Fig. 3b). We further explored the hypothesis that the mean primer parameters of all primer permutations would lie within our target range even if they were initially calculated based on the primer sequence including the most common nucleotides. We therefore computed the melting temperature, hairpin temperature, homodimer temperature, and GC content (Fig. 3e–h). In most cases, the means of primer permutations were within the target range or below the cutoff but showed a greater deviation from the optimum compared to the primer that was initially used for parameter calculation. The GC content is the least penalized parameter by varVAMP, and other parameters should have a more pronounced effect on primer selection in the current settings. Indeed, the GC content was also within the target range but more dependent on the MSA GC content (Fig. 3h).

The primary goal of varVAMP is to be aware of variations and to minimize primer mismatches with the input MSA sequences while maximizing amplicon coverage. To analyze whether varVAMP can outperform available software packages regarding variant-awareness, we also designed primers with both PrimalScheme and Olivar. To ensure a fair head-to-head benchmark, we aimed to design primers on the same input data using the same amplicon size constraints. However, Olivar requires a reference sequence and a mutation table, and PrimalScheme limits the number of sequences of the input MSA to 200. Therefore, we used consensus sequences and mutation frequency tables calculated from the respective gap-cleaned MSAs as the inputs for Olivar. For PrimalScheme, we constructed phylogenetic trees and used PARNAS[33] to subsample the MSAs for phylogenetic representative sequences. We also designed primer schemes with PrimalScheme on consensus sequences to compare the primer schemes to a control that lacked information about sequence variations (Table 2 and Supplementary Fig. 3a). Interestingly, both varVAMP and PrimalScheme consensus designs consistently maximized amplicon coverage. In contrast, Olivar and PrimalScheme MSA-based schemes often struggled to maximize 3' coverage but not 5' coverage, leading to lower predicted genome recovery. Next, we compared the mean mismatches of all primers compared to the input MSA (Table 2 and Supplementary Fig. 3b). While we did not observe apparent differences for the more conserved SARS-CoV-2 and BoDV-1 MSAs, varVAMP consistently had significantly lower mean primer mismatches for higher variable MSAs compared to Olivar and PrimalScheme designs. Interestingly, Olivar designs showed no significant mean mismatch differences to the PrimalScheme MSA designs with the only exception being ratHEV, which had the overall lowest mean sequence identity. Despite showing no clear difference to PrimalScheme in terms of variant awareness, Olivar had overall better running times than PrimalScheme MSA designs. However, compared with Olivar, varVAMP had running times 2x to 10x faster and typically finished within a few seconds (Table 2). Next, we compared the total number of mismatches of primers with the initial MSA. For varVAMP, most primers did not have multiple mismatches with MSA sequences for all designs (Fig. 4a). In comparison, both the PrimalScheme and Olivar designs had large proportions of primers with multiple mismatches. This was considerably more apparent for MSAs with higher variability, such as HEV or PV. varVAMP penalizes mismatches in the last five bases of a primer's 3' end to ensure stable target binding. By analyzing the position-dependent mismatches of all primers in the varVAMP schemes, we indeed observed that sequences of the input MSA displayed a consistently low frequency of mismatches at the 3' end of the primers with the most penalized last 3' prime position showing nearly no mismatches (Fig. 4b). For both the Olivar and PrimalScheme designs, mismatches at the 3' end were not preferentially minimized.

In conclusion, we designed primer schemes with varVAMP for different viruses with highly varying sequence variability. These

**Table 1 | Summary of varVAMP designs for tiled sequencing**

| Alignment statistics | | | | varVAMP parameters | | | | varVAMP output | | |
|---|---|---|---|---|---|---|---|---|---|---|
| Virus | Subtypes | n sequences | % mean sequence identity | Max ambig bases | Threshold | Optimal amplicon size | Maximum amplicon size | Expected recovery | n amplicons | varVAMP version |
| SARS-CoV-2 | B.1 - XBB | 865 | 99 ±1 | 1 | 0.99875 | 700 | 800 | 99.72% | 55 | v.0.9.4 |
| BoDV-1 | all | 55 | 89±8 | 2 | 0.94 | 400 | 550 | 98.6% | 27 | v.0.6 |
| HAV | all | 309 | 81±10 | 4 | 0.93 | 1000 | 1600 | 95.65% | 7 | v.0.8.3 |
| HEV genotype 3 | f, e | 376 | 76±6 | 4 | 0.91 | 1000 | 1500 | 99.02% | 7 | v.0.8.2 |
| HEV genotype 3 | c, h1, m, i, uc, l | 201 | 75±9 | 4 | 0.90 | 1000 | 1500 | 99.28% | 6 | v.0.8.2 |
| PV | 1-3 | 944 | 71±13 | 4 | 0.91 | 1000 | 1400 | 99.63% | 7 | v.0.8 |
| ratHEV | all | 41 | 57±10 | 5 | 0.82 | 1200 | 1700 | 97.41% | 6 | v.0.8.3 |

The table lists important alignment statistics, varVAMP input parameters and output information including the varVAMP version. Pairwise sequence identity was calculated with Identity (https://github.com/BioinformaticsToolsmith/Identity). Max. ambig. bases (-a parameter): Maximum number of ambiguous characters that are allowed within a primer sequence. varVAMP outputThreshold (-t parameter): Identity frequency threshold for a consensus nucleotide. All primers and their respective varVAMP outputs are accessible at: https://github.com/jonas-fuchs/ViralPrimerSchemes.

schemes had consistent primer statistics, maximized alignment coverage, and minimized mismatches to input MSA sequences with the most stringent constraints at the last position of the 3' end. In a direct head-to-head benchmark between the variant awareness of varVAMP, Olivar, and PrimalScheme, we show that varVAMP outperforms both software packages.

### Full-genome tiled amplicon sequencing of SARS-CoV-2, BoDV-1, HAV, PV and ratHEV

In a multicenter study with specialists for the respective pathogens, we evaluated whether the newly designed varVAMP primers for SARS-CoV-2, BoDV-1, HAV, PV and ratHEV were suitable for whole-genome sequencing and genome reconstruction. Similar to the HEV-3 primer schemes, we performed amplicon-based Illumina and, in the case of SARS-CoV-2 and some of the HAV samples, ONT sequencing on various samples via singleplex or multiplex PCRs. The sequencing protocols and selection of samples differed due to center-specific preferences. For SARS-CoV-2, we tested the novel primer scheme in multiplex PCR reactions on a random set of respiratory patient samples from circulating variants in early 2024 with different viral loads. Although some amplicons had lower coverage, we were able to construct complete genomes in the majority of cases (Fig. 5a and Supplementary Fig. 2). We evaluated BoDV-1 primers in multiplex reactions on three different virus stocks that had been isolated from brains of diseased patients in 2019, 2020 and 2022[34–36] and were cultivated on Vero cells. For all isolates, we were able to recover highly covered genome sequences (Fig. 5b and Supplementary Fig. 2). Only for the 2022 isolate, the poorer amplification of the last three amplicons led to slightly lower genome recovery (Supplementary Fig. 2). For HAV, we tested our HAV-specific primers on the cell culture derived laboratory strain V18-35519. Illumina sequencing yielded consistent and high coverage over all amplicons independent of multi- or singleplex reactions (Fig. 5c). Next, we transferred the protocol to three different HAV-positive patient samples: genotype IB-positive feces (patient 1) and sera (patient 3) as well as genotype IA positive feces (patient 2). Finally, we sequenced the sera of four additional patients via Oxford Nanopore: IA-positive (patient 4 and 5), IB-positive (patient 6) and IIIA-positive (patient 7). Full-genome recovery was achieved with all samples (Supplementary Fig. 2). Next, the PV primer scheme was tested on the Sabin 1-3 vaccine strains. Similar to previous results, sequencing resulted in high coverage and full-genome recovery. However, we observed that the third amplicon had overall lower coverage in multiplex but not in singleplex reactions (Fig. 5d and Supplementary Fig. 2). Finally, we evaluated the ratHEV primers that we designed to test the limits of varVAMP given the highest sequence variability and low number of MSA sequences (Table 1). We tested either single- or multiplex PCRs for the two previously described isolates R63 and pt2[37,38] (Fig. 5e). While we were able to achieve high coverage and genome recovery for the R63 isolate, we observed one and two amplicon dropouts for the pt2 single- and multiplex reactions, respectively (Supplementary Fig. 2).

We systematically evaluated the coverage and amplicon recovery for all primer schemes and samples (Fig. 6a). Most amplicons performed in a sample-dependent manner but in some cases multiplex performance was intrinsic to specific amplicons as exemplified by the third amplicon of the PV scheme (Fig. 5c). As all multiplex reactions across the tested schemes were performed with equimolar primer concentrations, we hypothesized that the performance could be improved by balancing primer concentrations As a proof-of-concept, we manually adjusted the molarities of the PV primers by increasing or decreasing them 0.25, 0.5 or 2 fold depending on the amplicons performance. Next, we re-evaluated the schemes uniformity by sequencing. After two consecutive rounds of manual primer adjustment, we achieved coverage for all Sabin strains that was comparable to the respective singleplex reactions (Supplementary Fig. 4). Analogous to the mismatch analysis with the input MSA (Fig. 4a), we also examined

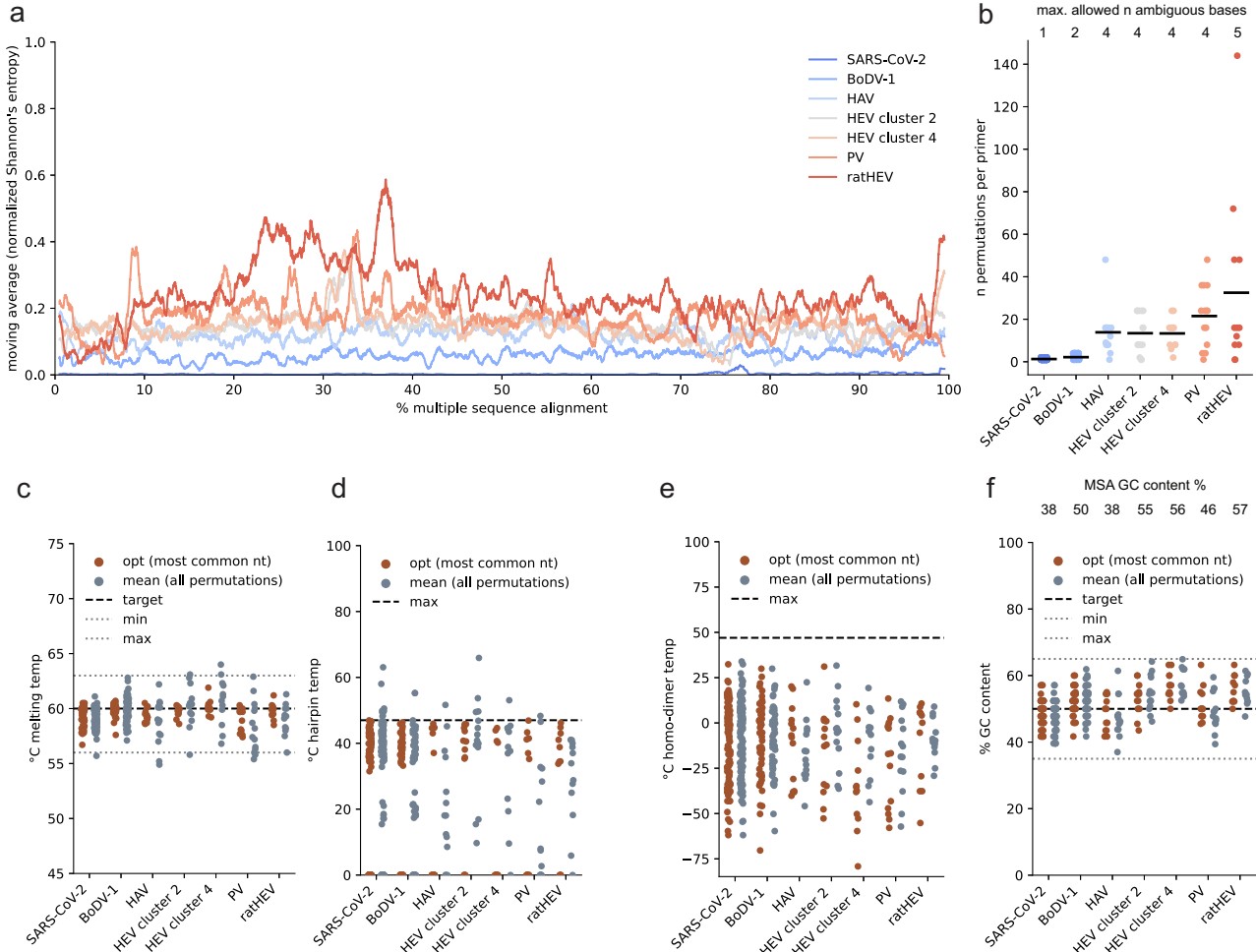

**Fig. 3 | In silico evaluation of novel tiled primer schemes for SARS-CoV-2, BoDV-1, HAV, HEV, PV and ratHEV. a** Normalized Shannon's entropy (1% rolling average) of the MSA used as the varVAMP input. **b** Number of permutations (degeneracy) of each primer in the tiled sequencing scheme for the respective viruses. Each dot shows the degeneracy of a single primer in the respective schemes (see Table 1). The horizontal lines indicate the means. (n number) Primer melting temperatures (**c**), hairpin temperatures (**d**), homo-dimer temperatures (**e**) and the GC content (**f**) were calculated either for the primer sequence including the most common nucleotides or averaged over all permutations of the final primer sequences that includes degenerate nucleotides. **e**, **f** were calculated with primer3. Each dot represents a single primer of the respective tiled schemes. The dotted lines indicate the upper target cutoffs or target ranges employed by varVAMP (nt nucleotide).

how many nucleotide mismatches between the primers and their target regions were present in our sequencing results (Fig. 6b). The primer target regions of the new sequences showed up to two mismatches to the degenerated primer sequences with the majority having no mismatches. Similar to the amplicon performance, the number of mismatches was mostly sample-dependent. We then explored if the three observed dropouts for HEV cluster 2 (patient 2), HEV cluster 4 (patient 4) and ratHEV (pt2) could have been caused by mismatches. Indeed, sequence analysis showed that in all cases one of the primers had two mismatches with one mismatch in the last five bases of the 3' end. In all cases, these variations were present in the initial alignment used for the primer design but not frequent enough to get integrated into the primer sequence (Supplementary Fig. 5). As the samples used for evaluation were selected in the respective centers based on availability and not sequence diversity, we tested whether this could have produced an unintended selection bias towards specific viral strains. Therefore, we evaluated whether the novel consensus sequences of each scheme have a variability that is comparable to that of the respective input alignments. The pairwise sequence identities of these small datasets were highly similar or lower to that of the alignment with only the newly produced sequences for HEV cluster 4 showing a significantly higher mean pairwise sequence identity of 7%

(Fig. 6c), indicating a slight bias to more highly conserved sequences. Lastly, we systematically analyzed off-target amplifications for all samples and schemes by analyzing unmapped non-viral reads with Kraken2 (Fig. 6d). We observed a high variability in the percentage of unmapped reads with most off-targets being classified as eukaryotic (human) or bacterial (various) (Supplementary Fig. 6). Interestingly, percentages of unmapped reads were not clearly correlated to the individual schemes and more dependent on the sample type. This was exemplified by the overall low percentage of unmapped reads for the BoDV-1 and PV schemes that were solely tested on cultured virus material and not on patient samples.

In summary, all primer schemes were suitable for tiled amplicon Illumina or ONT sequencing and resulted in highly covered full-genome sequences by applying center-specific sequencing protocols and bioinformatic pipelines initially developed for tiled amplicon sequencing of SARS-CoV-2.

## Design and wet-lab evaluation of PV qPCR primers designed with varVAMP
The WHO gold standard for PV detection is based on time- and resource-consuming virus cultivation. Molecular detection by qPCR is available but was designed for virus isolates propagated in cell

culture[39]. Moreover, primers and probes display a high level of degeneracy, decreasing the sensitivity of the assay and increasing the risk of unspecific non-viral amplification products for other sample types like wastewater. Therefore, we used varVAMP to design PV

serotype-specific assays. The optimal primer annealing temperature of each assay was tested with a gradient PCR ranging from 56–64 °C. All annealing temperatures resulted in the expected product size with 59 °C showing the lowest abundance of unspecific products. PV

**Table 2 | Summary of the software comparison between varVAMP, Olivar and PrimalScheme**

|  |  | SARS-CoV-2 | BoDV-1 | HAV | HEV cluster 2 | HEV cluster 4 | PV | ratHEV |
|---|---|---|---|---|---|---|---|---|
| varVAMP | Running time | 1 min 9 s | 16 s | 12 s | 22 s | 10 s | 11 s | 9 s |
|  | Number of amplicons | 55 | 27 | 7 | 7 | 6 | 7 | 6 |
|  | % alignment coverage | 100 | 99 | 96 | 99 | 99 | 100 | 97 |
|  | Mean mismatches per primer | 0.002 | 0.17 | 0.15 | 0.21 | 0.20 | 0.13 | 0.43 |
| Olivar | Running time | 14 min 54 s | 2 min 55 s | 38 s | 37 s | 39 s | 52 s | 28 s |
|  | Number of amplicons | 61 | 28 | 5 | 7 | 8 | 10 | 5 |
|  | % alignment coverage | 99 | 95 | 79 | 94 | 92 | 99 | 81 |
|  | Mean mismatches per primer | 0.001 | 0.44 | 0.88 | 1.03 | 1.09 | 1.63 | 2.11 |
| Primalscheme (alignment) | Running time | 7 min 32 s | 23 min 52 s | 17 h 11 min 33 s | 8 h 8 min 4 s | 50 h 25 min 56 s | 5 h 37 min 56 s | 48 s |
|  | Number of amplicons | 49 | 25 | 6 | 8 | 8 | 8 | 6 |
|  | % alignment coverage | 100 | 88 | 79 | 99 | 97 | 95 | 91 |
|  | Mean mismatches per primer | 0.027 | 0.29 | 0.78 | 1.04 | 1.19 | 1.65 | 4.17 |
| Primalscheme (consensus) | Running time | 8 s | 5 s | 6 s | 4 s | 4 s | 4 s | 4 s |
|  | Number of amplicons | 49 | 28 | 8 | 8 | 8 | 8 | 6 |
|  | % alignment coverage | 100 | 99 | 93 | 96 | 97 | 96 | 92 |
|  | Mean mismatches per primer | 0.027 | 0.73 | 1.63 | 1.48 | 1.63 | 2.60 | 3.04 |

Running times were analyzed on the same hardware (i7 1185G7 4×3 GHz, 32 GB DDR4 RAM, Ubuntu 22.04.3). Moreover, the number of amplicons, alignment coverage and mean mismatches with alignment sequences are summarized.

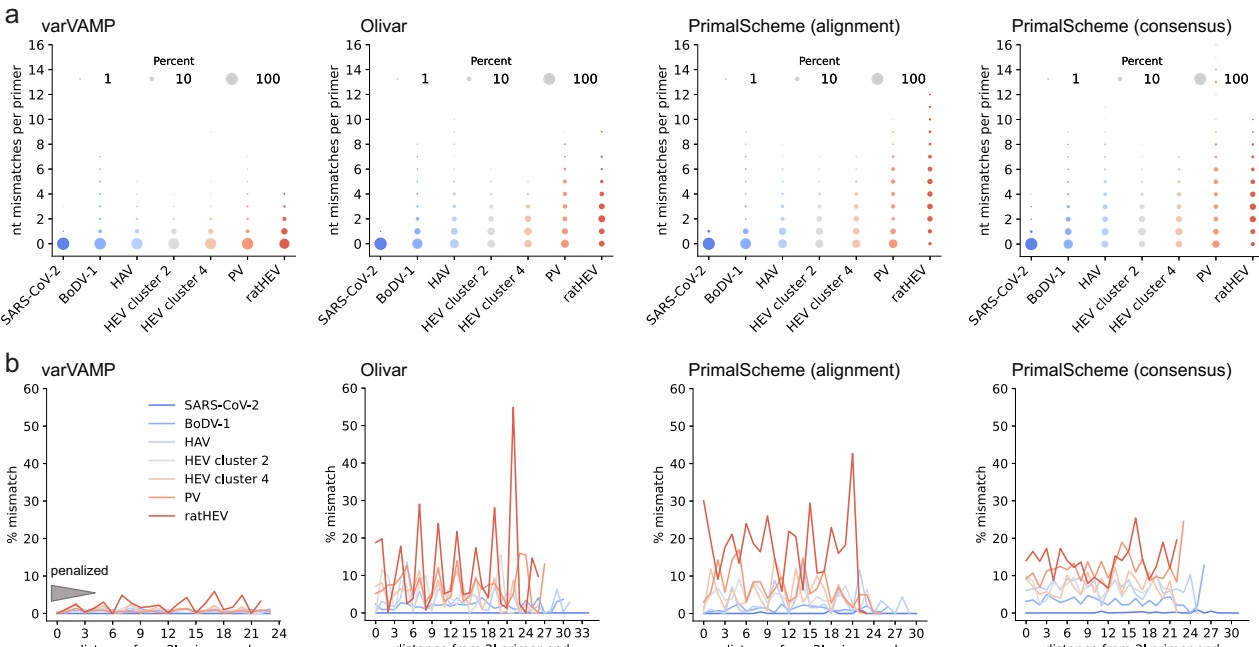

**Fig. 4 | Head-to-head benchmark of the variant awareness of varVAMP, Olivar and PrimalScheme. a** Cumulative counts of mismatches between primers and sequences in the MSA. For each primer the number of mismatches with each sequence of the MSA was counted if it was not covered by any primer permutation. Shown are the cumulative mismatches between primers and MSA sequences in the tiled primer schemes for the respective viruses. The dot area size is proportional to

the percentage. **b** Analogous to **a** the mismatches with the MSA sequences were counted per primer nucleotide position and averaged over all primers in a scheme. As primers vary in their length, the % mismatches are displayed starting at the primer's 3′ end (position 0 is the most 3′ nucleotide position). The gray triangle schematically indicates the primer positions that varVAMP penalizes and the position-specific penalty multipliers (32, 16, 8, 4, 2).

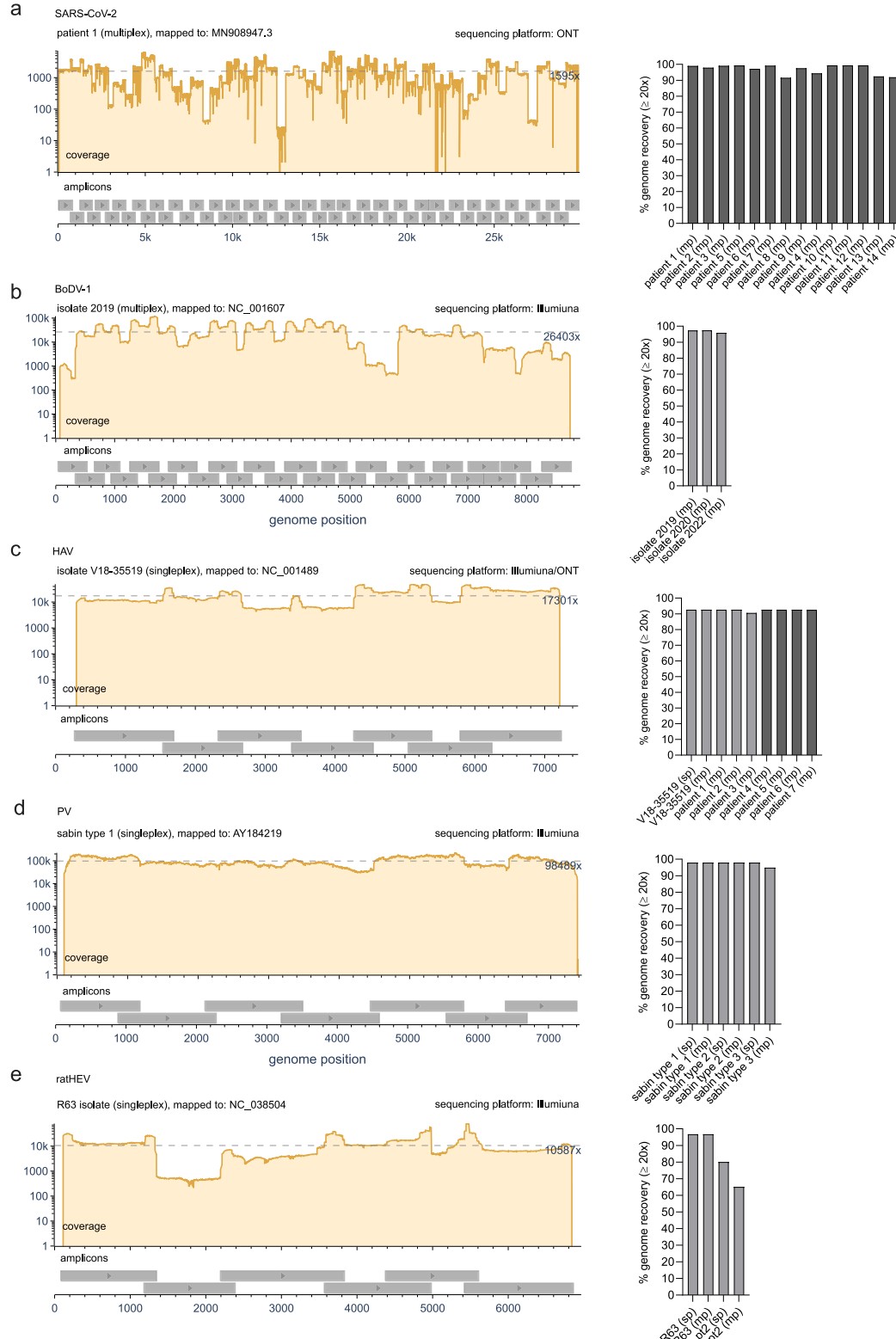

**Fig. 5 | Whole genome sequencing of SARS-CoV-2, BoDV-1, HAV, PV and ratHEV.** Representative coverage plots (left) and % genome recovery (right) of the different **a** SARS-CoV-2, **b** BoDV-1, **c** HAV, **d** PV and **e** ratHEV samples subjected to their respective tiled amplicon whole genome sequencing workflows. Coverage plots were created with BAMdash. The dotted lines indicate the mean coverages. The reference genomes used for mapping are indicated in the header of the coverage plots (individual coverage plots are given in Supplementary Fig. 2). Genome recovery was calculated as the precentage of reference nucleotides covered at least 20 fold (sp single plex, mp multiplex). Dark gray bars−ONT generated data, light gray bars−Illumina generated data.

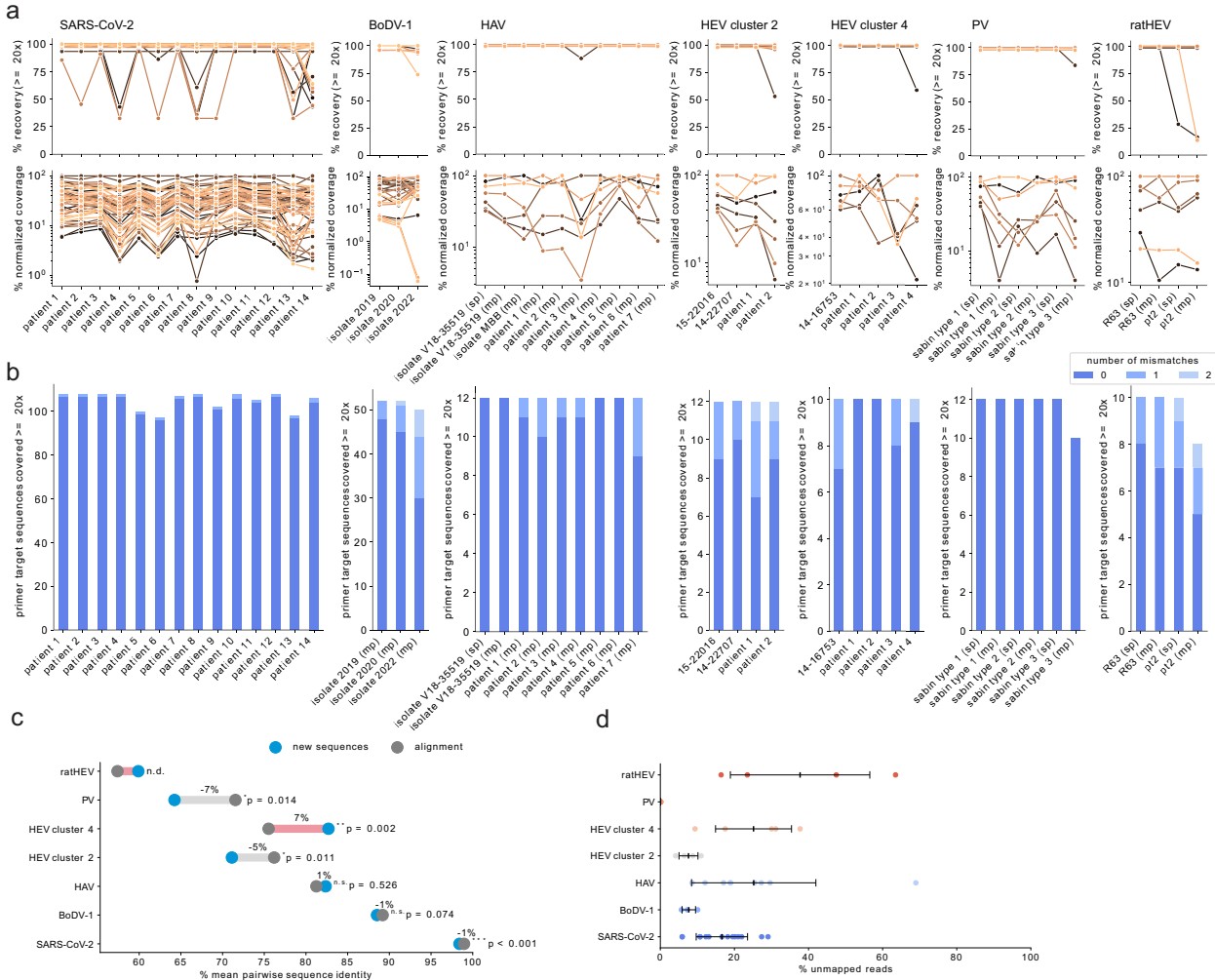

**Fig. 6 | Amplicon performance and mismatch analysis. a** For each sequencing result obtained via the virus specific primer scheme, the amplicon recovery (upper panel) and normalized coverage (lower panel) were calculated. Each color represents an individual amplicon for the respective schemes (see Table 1) tracked over different samples that are indicated on the *x*-axis. Amplicon recovery was calculated as the percentage of reference nucleotides covered at least 20-fold between the genomic start and stop position of the individual amplicons. For the normalized amplicon coverage, the mean coverage was calculated for each amplicon and normalized to the highest covered amplicon of each scheme (set to 100). **b** For each sequencing result, each primer binding region was analyzed for the number of mismatches not covered by any permutation of the corresponding primer sequence. Mutations were considered only if their variant frequency was ≥0.7. The primers were excluded from the analysis if any primer binding position was not covered at least 20-fold. **c** Dumbbell plot showing the pairwise identities of the newly generated fasta consensus sequences (blue dots) or the sequences of the varVAMP input MSA (dark gray dots) of each respective primer scheme. Light gray and red lines indicate the percent pairwise identity increase or decrease, respectively. Significance was calculated with a two-sided Welch's *t*-test between the pairwise identities of newly produced sequences and alignment sequences for each respective virus. Exact *p* values are given above the minimum significance threshold of 0.001. (n.d. not determined as $n < 3$, n.s. not significant, *$p \leq 0.05$, **$p \leq 0.01$, ***$p \leq 0.001$). **d** Percentage of off-target, unmapped reads for the respective schemes and samples (SARS-CoV-2: $n = 14$, BoDV-1: $n = 3$, HAV: $n = 9$, HEV cluster 2: $n = 4$, HEV cluster 4: $n = 5$, PV: $n = 6$, ratHEV: $n = 4$). Shown are mean ± STD.

serotype specific RT-qPCR assays were performed in a serial dilution experiment between $10^{-1}$ and $10^{-8}$ with RNA extracted from the viral supernatant. For all three types, PV detection was achieved up to a dilution of $10^{-7}$ (Fig. 7a–c). Absolute quantification analysis on the basis of dilution series yielded an efficiency value (PV1 = 1.88, PV2 = 1.87, PV3 = 1.90) close to 2 corresponding to a perfect amplification reaction. Cross-specificity testing revealed no detection between the Sabin strains indicating that the designed primers and probes are highly specific for their respective PV serotypes, despite primer and probe degeneration.

## Discussion

Here, we describe varVAMP, a command-line software tailored to pan-specific primer design for highly variable MSAs. Importantly, varVAMP is available through various bioinformatic repositories and has also

been deployed in Galaxy Europe (usegalaxy.eu), a web-based platform for bioinformatic data analysis[40]. On the basis of the input MSA, var-VAMP generates consensus sequences and analyzes them for the presence of potential primer sequences. From the subsequent pool of identified primers, varVAMP chooses optimal amplicons for specific molecular techniques such as tiled sequencing or qPCR.

varVAMP provides a fully automated primer design solution. However, a successful primer scheme can still face challenges. For example, selecting appropriate reference data prior to primer design can be difficult. In sequence data repositories, particularly in the case of viral sequences, there can be a lack of associated metadata, the presence of recombinant sequences, bias towards sequencing labs, geographic biases towards circulating strains, or an underrepresented amount of recently discovered or understudied viral pathogens[41,42]. Therefore, careful data selection is crucial for a successful primer

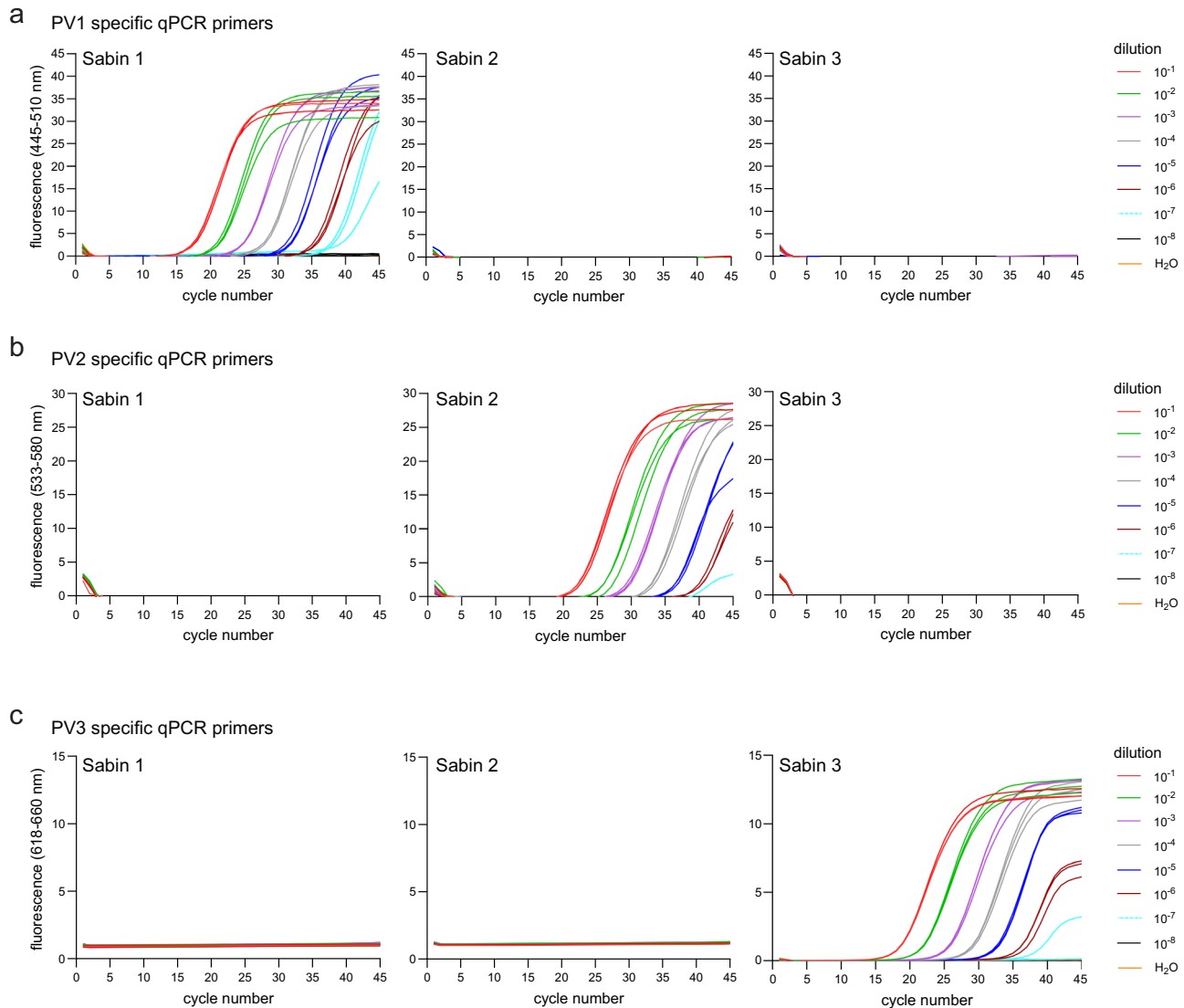

**Fig. 7 | Specificity and sensitivity of the novel PV qPCR schemes.** qPCR primers specific for **a** PV1, **b** PV2 and **c** PV3 were tested on serial RNA dilutions extracted from the viral supernatants of Sabin 1, Sabin 2 and Sabin 3 infected cell cultures ($n = 3$). The fluorescence was measured during the extension step with three different channel setups with respect to the probe fluorophore (FAM = 465–510 nm, JOE = 533–580 nm, CY5 = 618–660 nm). Amplification curves were analyzed using the Roche LightCycler 480 II device software.

design and might require the additional use of clustering algorithms and phylogenetic assessment tools[30,31,43,44]. The second challenge is that varVAMP parameters can be optimized depending on the input data. In particular the user-definable identity threshold and "number of allowed ambiguous bases" directly impact the ability of varVAMP to find primers. Notably, the software can ~~optimize~~ estimate one of the parameters by itself and comes by default with parameter definitions that should be broadly applicable. However, a certain degree of manual parameter adjustment, particular for the "number of ambiguous nucleotides" and the identity threshold, is always recommended to optimize the output.

We used varVAMP to design primers for tiled sequencing schemes of SARS-CoV-2, BoDV-1, HAV, HEV, PV and ratHEV and for qPCR of PV. To demonstrate its variant awareness, we also designed primer schemes with the software packages PrimalScheme and Olivar on similar input data and benchmarked them head-to-head to primers designed with varVAMP. We demonstrate that PrimalScheme and Olivar, to a similar degree, avoid designing primers in regions with high variability, thereby minimizing primer mismatches. However, for the genomes of more divergent pathogens such as HAV, HEV, and PV both

software packages were outperformed by varVAMP. The main reason is that instead of only avoiding sequence variations as best as possible, the rationale of our software is to incorporate variations as degenerate nucleotides into primer sequences. This massively increases its ability to compensate for mismatches with sequences of the initial alignment and potentially with novel sequences that harbor unknown combinations of these variations. Interestingly for higher variable data, both Olivar and PrimalScheme struggled to design amplicon schemes that maximized the 3' end, but not 5' end coverage of the reference sequence or alignment, respectively. Although speculative, this trend might result from the greedy or semi-greedy optimization amplicon search algorithms of PrimalScheme and Olivar, respectively. In contrast, varVAMPs graph-based approach is not greedy but finds the overall lowest penalized amplicon scheme while maximizing genome coverage. Another interesting finding of this comparison was that Olivar, specifically designed to incorporate information about variants in its primer selection, did not outperform the variant-awareness of PrimalScheme as previously published[20]. Both softwares clearly showed similar mean mismatches per primer with the input alignment and similar distributions of number of mismatches per primer again

strengthening the hypothesis that for higher variable alignments one can not simply avoid mismatches. It is, however, notable that Olivar has additional benefits, such as a BLAST module for avoiding off-target amplifications, sophisticated primer dimer considerations and drastically better running times compared to PrimalScheme.

varVAMP primer schemes perform well in real-world scenarios and with diverse sample types. Notably, we observed sample-dependent and -independent amplicon performance. The molecular reasons for sample-dependent PCR performance can be diverse. We show that primers lead to amplification even if there are mismatches between the primer and target sequence. However, if two or more mismatches are present, poorer amplification or complete dropouts could result. We hypothesized that mismatches were the likely cause of the HEV cluster 2, HEV cluster 4 and ratHEV amplicon dropouts. Indeed, in all cases the target sequence showed two mismatches with the primer sequence and a more detailed analysis revealed that one of these variations was always located in the last five bases of the 3′ end potentially affecting primer 3′ stability. Storage times or varying viral loads could have also impacted PCR amplification and off-target amplifications. Sample-independent amplicon performance is likely a PCR optimization issue. Compared with popular SARS-CoV-2 schemes[45], we have not optimized the PCR conditions, or primer pooling ratios. To provide an example of how to optimize the multiplex PCRs further, we adjusted the primer concentrations of the PV primers and showed that this led to a more balanced overall coverage.

All primer schemes described here have been developed because of a methodology gap. HEV phylogeny and sub-typing are mostly restricted to Sanger sequencing of ORF2 fragments thereby neglecting viral evolution in the remaining genome[46,47]. The current methodology to generating highly covered HEV full-genome sequences via Illumina or Oxford Nanopore requires costly RNA-Seq protocols using hybridization probes[48,49] or powerful sequencing machines. Our newly developed primer schemes could simplify sequencing and aid, for example, analyses of ribavirin resistance-associated mutations that can develop in immunocompromised HEV patients[50]. For PV, the molecular assays used for the detection and intratypic differentiation of serotypes are well suited for cell-culture isolates[51]. However, these techniques take time and effort. qPCR and amplicon-based NGS protocols for the rapid analysis of PV from patient and environmental samples are imperative for fast public health decisions. There are protocols for high-throughput sequencing of PV but similar to HEV they are restricted to a conserved part of the enterovirus genome[52]. Our novel methods for PV detection by qPCR and whole-genome sequencing could not only benefit existing surveillance programs but lay the foundation for wastewater surveillance strategies within the global PV eradication program[53,54].

Tiled primer scheme design with varVAMP has two main drawbacks. First of all, computation time is inversely correlated to alignment variability as higher conservation results in more found primers. More primers lead to higher computation due to the subsequent Dijkstra search as the algorithms time complexity is dependent on the number of edges and vertices in the amplicon graph. Therefore, varVAMP is likely computationally outperformed by both Olivar and PrimalScheme in the case for highly conserved alignments. Secondly, varVAMP performs a single round of primer swapping to solve heterodimers within pools which is not an optimal solution for highly multiplexed schemes as it can result in unsolvable heterodimer dimers and might require more than two primer pools to avoid all primer interactions. varVAMP analyses primer dimers on the basis of primer3 thermo-calculations and does not consider each primer permutation independently. These drawbacks in primer dimer considerations were required in our design approach as stringent primer dimer considerations would limit the pool of primers with minimal mismatches. The main hypothesis of varVAMP is that for higher variable alignments, mismatch minimization has to be prioritized to ensure primer-target

binding. This hypothesis was reinforced by our scheme evaluations as in all cases where we observed complete dropouts the target sequence had two rare variations in a primer binding region not covered by any primer permutation. Therefore, varVAMPs applicability is specifically tailored to higher variable and shorter viral genomes. For higher plexity schemes, primer dimer considerations grow more important due to the number of potential primer interactions. This problem has been e.g., addressed with the previously published SADDLE algorithm that is also used by Olivar[55]. Therefore, the choice of tiled primer design software relies on the variability of the input alignment and the intended plexity of the design.

In conclusion, we developed varVAMP for highly variable MSAs, for which it is nearly impossible to find primer sites that are well conserved. Instead of only minimizing mismatches, the software integrates variant information into primer sequences. The designed and validated primer schemes for the different viruses are a proof-of-concept for the applicability of varVAMP and have been developed because they could directly benefit viral diagnostics and epidemiology. Laboratories that have already established tiled amplicon sequencing pipelines or in-house qPCR protocols can adapt their methodologies to these new primer schemes with only minor modifications, while achieving a higher genome coverage, less amplicon dropouts, and overall higher-quality sequence data for diagnostics, research, and genomic surveillance.

## Methods
Research conducted with patient samples in this manuscript complies with relevant ethical regulations. The ethical oversight depended on the respective centers. Next generation sequencing was performed in the Diagnostic Department of the Institute of Virology, University Medical Center, Freiburg using pseudonymized patient left over specimens from HEV positive patients (Local ethics committee no. 1001913). For HAV patient samples, the molecular diagnostics were collected and analyzed on the basis of the German Infection Protection Act. Thus, a review by an ethics committee was not required. Analyses of SARS-CoV-2 positive samples collected within the IMSSC2 network were conducted according to the principles laid down in the Helsinki Declaration with anonymized data.

### Software
varVAMP is an easy-to-use cross-platform command line tool that is available via PyPI, DOCKER, BIOCONDA, and the Galaxy platform. It enables primer design for a variety of molecular techniques, including single amplicons, tiled full genome sequencing, and qPCR. In the following, we describe the different algorithms and steps that have been implemented for varVAMP. The primer design pipeline only requires an already computed multiple sequence alignment as input. Importantly, all the parameters are highly customizable via direct arguments or a config file. varVAMP performs the following major steps (Fig. 1a and Supplementary Fig. 1): (1) alignment masking, (2) consensus sequence generation, (3) potential primer region search, (4) evaluation of suitability of digested k-mers for primers or qPCR probes, and (5) amplicon scheme creation. The last step differs depending on the three different modes available for varVAMP: single, tiled, and qPCR. To evaluate potential off-target amplifications, varVAMP can use a BLAST database to predict off-targets and preferentially selects amplicons without BLAST matches in the final amplicon scheme.

In the first step, varVAMP can estimate some of the user-parameters. For a minimal primer length $l_{min}$, two main parameters influence primer design: number of ambiguous nucleotides tolerated within a primer sequence $n_a$ with $n_a \in \mathbb{N}$, $0 \leq n_a \leq l_{min}$ and the identity threshold for a nucleotide $t$ to be considered a consensus nucleotide with $t \in \mathbb{R}$, $0 \leq t \leq 1$. Parameter estimation is only performed for either $n_a$ or $t$. If both $n_a$ and $t$ are not given, varVAMP sets a default value for $n_a = 2$. For each iteration of the parameter testing, either $n_a$ is

incremented by −1 starting at the maximum primer length or $t$ is incremented by 0.01 starting at 0.1. After incrementing the non-fixed parameter, the highest nucleotide frequency at each alignment position is determined. Next, the lengths of non-overlapping nucleotide stretches that consist of nucleotides reaching the current threshold $\{l_1, l_2 \ldots l_m\}$ are calculated. The coverage $c$ of the given alignment with the length $l_a$ that can be considered for potential primer regions is then estimated as follows:

$$c \approx \frac{\sum_{L_i \in L} L_i}{l_a}, \quad L_i = \sum_{k=i}^{i+n_a} l_k, \; L = \{L_i | L_i \geq l_{\min}\} \quad (1)$$

Based on empirical testing, we found that estimates are close to manually optimized if 50% of the alignment can be considered for potential primers. Therefore, parameter testing is stopped when in an iteration $c \geq 0.5$. If varVAMP is used to design qPCR schemes, the number of ambiguous characters for the qPCR probe $n_p$ is set to $n_p = n_a - 1$ to ensure greater specificity of the probe than the flanking primers. Notably, the automatic parameter estimation will not produce optimal values for all alignments but will give a reference point on how to set the $n_a$ or $t$ parameters. We always recommend manual parameter tweaking for $n_a$ or $t$.

In the following alignment preprocessing step, all characters are converted to lowercase. Next, gaps in the alignment are masked. Common gaps are defined as gaps in a higher frequency than $1 - t$ as these positions have not enough nucleotide information to reach the desired $t$. Common gaps are then masked with 'N' (single nucleotide deletions) or 'NN' (larger deletions) in the multiple sequence alignment. This ensures that primers will not span regions that could be potential INDEL sites and that for the large majority of sequences, the amplicon size is not overestimated. Moreover, qPCR amplicons are not considered if they would span large deletions as small deviations in amplicon length are particularly problematic for smaller qPCR amplicons.

In the next step, two consensus sequences are deduced from the gap-masked alignment. At each alignment position, the sorted list of observed nucleotide frequencies is calculated. If a sequence in the alignment contains ambiguous nucleotides, all permutations of these nucleotides are considered and added to the nucleotide frequencies proportionally to the number of permutations. The first consensus sequence is generated simply from the most frequent nucleotide at each site. This majority consensus sequence is the basis for the primer search. For the second consensus sequence, the observed nucleotide frequencies at each site are added, starting from the highest frequency, until their sum reaches $t$. The IUPAC symbol for the set of nucleotides that contribute to the frequency sum is then taken as the consensus at the site. This symbol will be identical to the corresponding nucleotide in the majority consensus if the frequency of that nucleotide alone reaches or exceeds $t$. This second consensus sequence allows the search for regions that have only a certain number of ambiguous characters within the minimal primer length.

Next, the consensus sequence with ambiguous nucleotide characters is searched for potential primer regions. The algorithm opens a region window at the start of the sequence. The window is closed if $>n_a$ ambiguous nucleotides are found within a sequence of $l_{\min}$ nucleotides or a gap is reached. Importantly, the algorithm is case sensitive which allows the differentiation between 'N'/'NN' (gaps) and 'n' (any nucleotide) characters. We define the resulting window as $w = [w_{\text{start}}, w_{\text{end}}]$ with ambiguous character positions $x_1, \ldots, x_m$. If the window is closed due to a gap, a new window is opened at the subsequent nucleotide $w_{\text{end}} + 1$. If the window was closed because of the number of ambiguous characters, the new window is opened at the position after the first ambiguous character counting towards $n_a$ that led to closing the previous window, $x_1 + 1$. Regions are considered for the primer search if $w_{\text{end}} - w_{\text{start}} \geq l_{\min}$.

In the identified primer search regions, the majority consensus is digested into all possible unique $k$-mers for $l_{\min} \leq k \leq l_{\max}$. Each k-mer is subsequently evaluated for its suitability as a primer. For this purpose, primer3[17] is used and some of the rationales and functions were adapted from PrimalScheme[10]. First, each k-mer is hard-filtered independent of its direction for unacceptable temperature, size, GC content, homopolymer length, di-nucleotide repeats, and homodimer formation. Importantly, varVAMP filters primer dimers (homo- and hetero-dimers) based on a user-definable melting temperature cutoff and thermo-calculations with primer3. Similar to PrimalScheme, a base penalty $p_b$ for the k-mers' deviations from the optimal temperature, size and GC content is calculated and also hard-filtered if it exceeds the base penalty threshold. The primers are subsequently evaluated for their suitability as forward or reverse primers by filtering out unacceptable hairpin formation temperatures, the 3-prime presence of ambiguous characters, and the absence of a GC clamp. For primers surviving all filtering steps, a permutation penalty $p_p$ and a 3' mismatch penalty $p_m$ are calculated. $p_p$ is calculated as the number of primer permutations of the primer version that has ambiguous characters (deduced from the ambiguous consensus sequence) multiplied by the user-definable permutation penalty multiplier. For a primer with $n_p$ characters, we define position-specific penalties $\{c_1, c_2, \ldots c_{n_p}\}$ and calculate the mismatch frequencies at each position $\{f_1, f_2, \ldots f_{n_p}\}$. We then calculate:

$$p_m = \sum_{i=1}^{n_p} c_i \cdot f_i \quad (2)$$

Note that only the last five positions receive non-zero multipliers in the standard settings with the highest multiplier at the very 3' position to enforce stable binding of the 3' end. The final primer penalty $p_{\text{primer}}$ is then calculated as:

$$p_{\text{primer}} = p_b + p_m + p_p \quad (3)$$

$p_{\text{primer}}$ shows the primer deviations from the base parameters, the number of permutations and mismatches at the 3'-prime end. The closer $p_{\text{primer}}$ is to zero, the better it represents an optimal primer.

To reduce the number of potential primers, all primers are penalty-sorted from low to high. From this sorted list, primers are retained if they do not overlap with the middle third of an already retained lower scoring primer, improving the final selection of primers with minimal overlap and minimal $p_{\text{primer}}$. Next, all potential non-dimer forming combinations of forward and reverse primers within a given amplicon range length are computed. A resulting amplicon $a_i$ is defined by the primers $\text{primer}_{fw}$, $\text{primer}_{rv}$. Given the length of the amplicon $l_i$ and the user-defined optimal amplicon length $l_{opt}$, we define the amplicon penalty $p_i$:

$$p_i = (p_{\text{primer}_{fw}} + p_{\text{primer}_{rv}}) \cdot e^{\frac{l_i}{l_{opt}}} \quad (4)$$

This ensures that the amplicon selection is length dependent and that amplicons with a length closer to the optimal amplicon length are favored. For a single-amplicon design, amplicons are sorted by their penalty from low to high, and only low-scoring non-overlapping amplicons are retained.

For the tiled approach, a weighted directed graph $G = (V, E)$ with vertices $V$ and edges $E$ is created. Each vertex $v_i \in V$ represents an amplicon $a_i$ and the set of vertices is given by $V = \{v_1, v_2, \ldots v_m\}$. We define the vertex start $\text{start}_{v_i}$ as the position of the first nucleotide in $\text{primer}_{fw}$ belonging to $a_i$ and the vertex stop $\text{stop}_{v_i}$ as the position of the last nucleotide in $\text{primer}_{rv}$ belonging to $a_i$. An edge $e \in E$ is defined as a tupel $v_i, v_j, w_j$ of two distinct nodes.

The edge weight $w_j = (o_j, p_j)$ incorporates information about whether amplicon $a_j$ generates an off-target hit with the optional

BLAST database and amplicon penalty $p_j$. If an off-target is generated $o_j$ is 1 otherwise it is 0. The set of all edges $E$ is defined as:

$$E = \{(v_i, v_j, w_j) | v_i, v_j \in V \text{ and } i \neq j \text{ and } v_i, v_j \text{ overlap}\} \quad (5)$$

We say that $v_i, v_j$ overlap if they satisfy the user-defined reciprocal pairwise sequence overlap between the 5' end position of $\text{primer}_{rv}$ of $v_i$ and 3' end position of $\text{primer}_{fw}$ of $v_j$. Moreover, the $\text{start}_{v_j}$ cannot be located in the first half of $v_i$. Next, varVAMP searches for the shortest path in $G$ from a source vertex $v_s$ with Dijkstra's algorithm[25]. The stop position with the highest genomic index of all $v_i \in V$ is denoted $\text{stop}_{\max} = \max\{\text{stop}_{v_i} | i = 1, ..., m\}$ and the lowest penalized amplicon with the farthest stop position reached by Dijkstra's search is termed $v_{\max}$. The amplicon coverage over the consensus sequence is defined as:

$$c_{con} = \text{stop}_{v_{\max}} - \text{start}_{v_s} \quad (6)$$

We store the current highest coverage $c^*_{con}$, and Dijkstra's search is repeated for all $v_i$ until $\text{start}_{v_i} + c^*_{con} > \text{stop}_{\max}$. Therefore, the shortest path resulting in the highest coverage is the path that results in $c^*_{con}$. varVAMP defines two amplicon pools containing non-adjacent amplicons for the final scheme to allow primer multiplexing. In the last step, both pools are analyzed for the presence of primer heterodimers. If heterodimers are found, varVAMP considers the previously excluded primers overlapping with the middle third of the heterodimer-forming pair and tries to find primers that do not form heterodimers within the respective primer pools. This heterodimer solve is only performed once and subsequent new heterodimers are reported as unsolvable.

For the qPCR mode, the consensus sequence containing ambiguous nucleotides is searched for regions that satisfy the specific length and $n_p$ constraints of the qPCR probe and is again digested into all possible unique k-mers within the probe size range. These k-mers are tested and evaluated for their suitability as qPCR probes in a manner analogous to primer screening. However, here we apply additional constraints: (1) probes are not allowed to have ambiguous bases at either end, (2) probes cannot have a guanine at the 5' end as this might result in quenching, and (3) their direction is defined so that the qPCR probes have more cytosines than guanines. Next, varVAMP searches for potential qPCR amplicons. This is achieved by searching for primer subsets within the amplicon length constraint flanking a qPCR probe. Potential amplicons are excluded if they violate the GC content constraint or contain large deletions. The flanking primers must be within a narrow temperature range, the probe must have a higher temperature than the primers, they cannot form dimers with each other, and the probe must be within a certain distance to the primer on the same strand. varVAMP also evaluates the presence of dimers in all probe-primer permutations and excludes primer-probe combinations that overlap at their ends, as this might also lead to unspecific probe hydrolysis. Finally, amplicons are sorted by their amplicon penalty:

$$p_i = p_{\text{primer}_{fw}} + p_{\text{primer}_{rw}} + p_{\text{probe}} \quad (7)$$

Afterwards amplicons are tested for their $\Delta G$ at the lowest primer temperature using seqfold (https://github.com/Lattice-Automation/seqfold). varVAMP reports amplicons that pass the $\Delta G$ cutoff.

varVAMP has an optional BLAST feature that allows evaluation if amplicon primers could result in off-target products via custom BLAST database[56]. Here, we perform a relaxed BLAST search with the BLAST settings published for primerBLAST[57]. Afterwards, the results are filtered for matches with a user-definable minimal overlap of identical nucleotides while considering both query coverage and mismatches. We now check each amplicon for potential off-target hits defined as matches for both primers that are sufficiently close together on the same reference sequence, but on opposite strands. Amplicons that result in off-target hits, are preferentially not considered in the final

scheme. In the single and qPCR modes, amplicons are first sorted for the absence of off-targets and then by their penalty. In tiled mode, the edge weights of the vertices include both off-target information and the amplicon penalty. Here, the shortest path is first evaluated on the amount of off-target hits generated by the path before considering the cumulative amplicon penalty, thereby avoiding, but not excluding amplicons with off-target effects.

The final primers (independently from the varVAMP mode) are deduced from the consensus sequence incorporating degenerate nucleotides.

## Primer design via varVAMP

Data selection for the multiple sequence alignments (MSAs) that were used as the inputs for varVAMP was highly dependent on the individual pathogens.

For SARS-CoV-2, we obtained 920,323 full-length genome sequences, sampled between 2021-10-11 and 2023-09-26, and their lineage assignments from https://github.com/robert-koch-institut/SARS-CoV-2-Sequenzdaten_aus_Deutschland (accessed 2023-10-13). Covsonar (https://github.com/rki-mf1/covsonar, v1.1.9) was used to calculate mutation profiles for all sequences, followed by a Python script (https://github.com/rki-mf1/sc2-mutation-frequency-calculator, v0.0.2-alpha) to select characteristic mutations per lineage (75% frequency). The script then uses these characteristic mutations to construct a single, representative consensus sequence per lineage. A representative consensus sequence was only calculated if at least ten genomes were available for a particular lineage. This resulted in representative consensus genomes for 865 SARS-CoV-2 lineages which were then aligned and used as input for varVAMP.

For BoDV-1 we downloaded all available full-length sequences that belong to the *Orthobornavirus bornaense* species (BoDV-1: 54 sequences, BoDV-2: 1 sequence). For HAV we downloaded all available full-length sequences that belong to the *Hepatovirus A* species (326 sequences). Patent and artificial clone sequences were excluded resulting in 309 HAV sequences. PV sequences were filtered in a similar manner and we excluded, by manual alignment inspection, highly divergent sequences that were likely the result of recombination events with other Enteroviruses yielding 944 sequences. For the qPCR designs, we split this dataset, on the basis of metadata, into the individual serotypes 1–3 resulting in 241, 494 and 209 sequences, respectively. For HEV data selection, we downloaded all available full-length sequences of the *Hepeviridae* family (1377 sequences). Patent and artificial clone sequences were excluded resulting in 1349 sequences. The remaining sequences were compared with the HEV reference set by Smith et al.[58], which was extended with the reference sequences for rat, bird, bat, fish, frog and planthopper HEV (NC_038504.1, NC_023425.1, NC_018382.1, NC_015521.1, NC_040835.1 and NC_040710.1, respectively), via the ggsearch36 algorithm[59]. This classification resulted in 1222 HEV sequences and 71 ratHEV sequences. Next, we used the greedy clustering algorithm of vsearch 2.22.1[30] with global clustering thresholds of 0.82 and 0.71, respectively, to further split the HEV and ratHEV datasets by similarity. Clustering results were manually inspected in phylogenetic trees constructed with IQ-TREE 2 under the GTR + F + R10 substitution model and 1000 bootstrap replicates[31]. For HEV we choose two clusters that reflect the most common European HEV-3 subgenotypes (HEV-3 f, e and HEV-3 c, h1, m, i, uc, l). For ratHEV we focused on the largest cluster essentially excluding ratHEV from non-rat species and further excluded sequences that were too short resulting in a total of 41 sequences.

Next, the pairwise sequence identity within each sequence batch was calculated with Identity[60] and the sequences were aligned with MAFFT[32] with default settings. These alignments were then used as the input for varVAMP. On the basis of sequence identity, we chose, for tiled sequencing, to fix the allowed maximum number of ambiguous characters within the minimum primer length depending on the mean

sequence identity within the batch ($n_a = 2$ with 90% identity, $n_a = 4$ between 70 and 80%, $n_a = 5$ below 70%). Next, the identity threshold $t$ was maximized until varVAMP could not find a tiled scheme that covered the whole genome (Table 1). For the qPCR designs we chose to allow one less ambiguous base for the probe than for the primers (Table 3). Here, settings were selected on the basis of whether var-VAMP was able to find a qPCR scheme under the $\Delta G$ constraints rather than solely on sequence similarity. All input alignments and varVAMP outputs are available at: https://github.com/jonas-fuchs/ViralPrimerSchemes.

### Primer design via PrimalScheme and Olivar

To achieve a fair head-to-head comparison of varVAMP to Pri-malScheme and Olivar, we designed primer schemes on the basis of similar sequence information and the same constraints on ampli-con sizes.

Olivar does not utilize alignments. Instead, it uses a reference sequence and a mutation table. Therefore, we used a gap-cleaned alignment and determined all nucleotide variations with SNP-sites v.2.5.1[61]. We then used a custom Python script (available at https://github.com/jonas-fuchs/varVAMP_in_silico_analysis) to convert the output of SNP-sites to an Olivar compatible variant table and generate a consensus sequence that includes the most frequent nucleotide at a given position as reference. The primers were then designed with Olivar v1.1.5 using the consensus sequence and mutation table.

For PrimalScheme, we designed two separate amplicon schemes. The first was based on the respective consensus sequences of each alignment, which did not include information about variations at respective sites. The second batch was based on the alignments. As PrimalScheme v.1.4.1 limits the number of sequences in an alignment to 200, we generated a phylogenetic tree with FastTree2 v.2.1.11[62] and used PARNAS v.0.1.5[33] to down sample the alignments for SARS-CoV-2, HAV, HEV, and PV while keeping representative sequences. For HEV clusters 2 and 4, we had to down sample to 100 sequences due to running time and RAM constraints, as the software crashed on multiple hardware configurations with more sequences. We further excluded short sequences (HAV: MW405350, AY974170, OQ077994, MF621615; HEV cluster 2: MN614143; ratHEV: KY432898). The down sampled alignments or the alignments for BoDV-1 and ratHEV were then used as the inputs for PrimalScheme. Finally, for a more straightforward ana-lysis, we manually adjusted the BED file primer positions to match the respective positions in the alignment rather than those provided by PrimalScheme.

All software was run on the same hardware (i7 1185G7 4×3 GHz, 32 GB DDR4 RAM, Ubuntu 22.04.3). The running times of all three soft-ware packages were extracted or calculated from the representative console output (Olivar) or log files (varVAMP and PrimalScheme).

### HEV qPCR and HEV sub-genotyping

Patient serum samples were tested for HEV via the HEV RT-PCR Kit 1.5 (AS0271543, Altona Diagnostics, AltoStar®). For HEV sub-genotyping of HEV-positive samples, we used an in-house nested RT-PCR (210212, Qiagen, Hilden, Germany) protocol based on previously published primers and nested primers in a conserved region of ORF1[63]. RT-PCR was performed at 42 °C for 60 min, 15 min at 95 °C followed by 40 cycles at 94 °C (30 s), 56.5 °C (30 s) and 74 °C (45 s). The final elonga-tion was at 74 °C for 5 min. Agarose gel negative PCRs were subjected to a nested PCR. Afterwards, the PCR products were Sanger sequenced.

### Production of virus stocks

The BoDV-1 strains were derived from native human brain sections[34–36] and propagated in Vero cells (ATCC: CCL-81) with DMEM supple-mented with 10% heat-inactivated fetal calf serum (FCS), 90 U/ml streptomycin, 0.3 mg/ml glutamine, and 200 U/ml penicillin (PAN Biotech, Aidenbach, Germany). Permanently infected cells were split twice a week and monitored for *Mycoplasma* spp. contamination every 12 weeks. To obtain a cell-free viral stock, the cell culture supernatants were centrifuged at 1000 rcf to remove cell debris and filtered through Rotilab syringe filters with a pore size of 0.22 μm (Carl Roth, Karlsruhe, Germany).

HEV-containing supernatants were harvested from persistently infected cell cultures. The liver carcinoma cell lines (PLC/PRF/5, ATCC: CRL-8024) persistently infected with HEV-3c strain 14-16753, HEV-3e strain 14-22707 or HEV-3f strain 15-22016 (provided by the National Consultant Laboratory for HAV and HEV, University Hospital Regens-burg) were maintained in modified Minimum Essential Medium at 37 °C and 5% $CO_2$[29].

HAV genotype IB strains MBB[64] and V18-35519 (derived from the plasma of a patient with acute hepatitis A) were both propagated in HuH-7 cells maintained in BMEM and incubated at 34.5 °C and 5% $CO_2$. The MBB and V18-35519 strains were harvested at 665 and 378 days post inoculation, respectively.

RD-A cells were infected with PV for virus propagation and culti-vated in MEM Earls media supplemented with L-glutamine, 1x non-essential amino acids, 100 U/ml penicillin and streptomycin (61100087, 11140050, 15140122, Thermo Fisher Scientific, Germany) and 7.3% heat inactivated fetal calf serum (BioWest, South America). RD-A cells were split once a week, and an internal quality control was performed at passage five to ensure cell sensitivity. The cell culture was conducted at 37 °C and 5% $CO_2$. Cultures were checked daily for a cytopathic effect (CPE) for a maximum of 7 days. Up to 2 ml of cell cultures with an observed CPE were centrifuged for 4 min at 1400 × $g$, and the supernatant was used as the viral stock.

The ratHEV strain R63[37], which was originally detected in a Norway rat from Germany, and the ratHEV strain pt2[38], which was identified in a human patient in Hong Kong, were generated and propagated in HuH-7-Lunet BLR cells under conditions as described previously[65,66]. The ratHEV positive culture supernatants were harvested at 66 days post infection.

### Illumina sequencing of HEV-3

Viral RNA was isolated via the QIAamp® Viral RNA kit (52904, Qiagen, Hilden, Germany) following the manufacturer's protocol. A one-step RT-PCR using SuperScript™ IV One-Step RT-PCR System (12594025, Thermo Fisher) was subsequently performed for each amplicon

**Table 3 | Summary of varVAMP designs for qPCR**

| Alignment statistics | | | | varVAMP parameters | | | | varVAMP output | |
|---|---|---|---|---|---|---|---|---|---|
| Virus | Subtypes | *n* sequences | % mean sequence identity | Primer max ambig bases | Probe max ambig bases | Threshold | ΔG cutoff (kcal/mol) | *n* found schemes | varVAMP version |
| PV | 1 | 241 | 86 ± 12 | 2 | 1 | 0.93 | −3 | 3 | 0.7 |
| PV | 2 | 494 | 88 ± 8 | 1 | 0 | 0.98 | −3 | 4 | 0.7 |
| PV | 3 | 209 | 91 ± 9 | 2 | 1 | 0.93 | −3 | 3 | 0.7 |

Important alignment statistics are listed together with varVAMP input parameters and output information, including the varVAMP version. Pairwise sequence identity was calculated with Identity (https://github.com/BioinformaticsToolsmith/Identity). All primers and their respective varVAMP outputs are accessible at: https://github.com/jonas-fuchs/ViralPrimerSchemes.

separately, with a total primer concentration of 1 μM. To reduce non-specific amplification, reverse transcription was performed at 55 °C for 60 min and we gradually reduced the primer annealing temperature during the first 10 PCR cycles (10 s at 98 °C, 10 s at 63 °C (−0.5 °C/cycle), 2 min at 72 °C) and then performed another 35 cycles at a constant annealing temperature (10 s at 98 °C, 10 s at 58 °C, 2 min at 72 °C). Next, amplicons were pooled and purified with AMPure XP beads (A63881, Beckman Coulter). A total of 50–100 ng DNA was prepared for Illumina sequencing using the NEBNext Ultra II FS DNA Library Prep Kit (E6177, NEB, Frankfurt am Main, Germany). Normalized and pooled sequencing libraries were denatured with 0.2 N NaOH and sequenced on an Illumina MiSeq instrument (300-cycle MiSeq Reagent Kit v2, MS-102-2002, Illumina).

### Illumina sequencing of BoDV-1, HAV, ratHEV

Viral RNA was isolated using the QIAamp Viral RNA Mini Kit (Qiagen, Hilden, Germany) or the EMAG Nucleic Acid Extraction System (Biomeriéux Deutschland GmbH, Nürtingen, Germany). The RNA was transcribed into cDNA with LunaScript RT SuperMix Kit (New England Biolabs, Ipswich, MA, USA) for 2 min at 25 °C, 10 min at 55 °C and 1 min at 95 °C. The cDNA was then amplified in single reactions with primer pairs, as well as multiplex PCRs with primer pools using the Q5 Hot Start-Fidelity DNA Polymerase Kit (New England Biolabs, Ipswich, MA, USA) with an initial step at 98 °C for 30 s, followed by 35 cycles (15 s at 98 °C, 5 min 65 °C) and a final extension step at 65 °C for 5 min. Illumina sequencing was performed analogous to the HEV-3 sequencing protocol.

### Illumina sequencing of PV

PV vaccine strain 1-3 (Sabin 1-3) RNA was isolated from infected RD-A cells using the QIAamp® Viral RNA kit (52904, Qiagen, Hilden, Germany) following the manufacturers' protocol, with PV2 being archived RNA for containment reasons. A one-step RT-PCR using the Qiagen One-step RT-PCR kit (210212, Qiagen, Hilden, Germany) was subsequently performed for each amplicon separately or for the respective pools with a total primer concentration of 0.6 μM. The reverse transcription was performed at 50 °C for 30 min followed by an initial PCR activation step at 95 °C for 15 min. Amplification was performed for 40 cycles with a stepwise reduction in primer annealing temperature during the first 10 cycles (30 s at 94 °C, 45 s at 70 °C ($\Delta T$ −1 °C/cycle), 90 s 72 °C) and a constant annealing temperature for the next 30 cycles (30 s at 94 °C, 45 s at 60 °C, 90 s 72 °C) and a final extension step for 10 min at 72 °C. Multiplex RT-PCR pools of each sample were combined and purified via MagSi-NGSPREP-PLUS beads (MDKT00010075, Steinbrenner, Germany) according to the manufacturer's manual and DNA concentration measured using the Qubit™ 1X dsDNA Assay-Kit (Q33230, Thermo Fisher Scientific, Germany). For Illumina sequencing, a library was prepared using 70–400 ng DNA with the Nextera XT DNA Library Preparation Kit (FC-131-1096, Illumina) and sequenced on an Illumina MiSeq Instrument (2 × 300 bp read length).

### Tiled ONT sequencing for HAV

Nucleic acid was extracted from samples on an EZ1® Advanced XL workstation using the EZ1 Virus Mini Kit v2.0 (Qiagen, Hilden, Germany) and transcribed into cDNA with LunaScript RT SuperMix Kit (New England Biolabs, Ipswich, MA, USA) for 2 min at 25 °C, 10 min at 55 °C and 1 min at 95 °C. The cDNA was then amplified in multiplex PCRs with HAV-specific primer pools using the Q5 Hot Start-Fidelity DNA Polymerase Kit (New England Biolabs, Ipswich, MA, USA) with an initial step at 98 °C for 30 s, followed by 35 cycles (15 s at 98 °C, 5 min at 65 °C) and a final extension step at 65 °C for 5 min. Barcoding was performed with eight samples per run using the Rapid Barcoding Kit 96 V14 (Oxford Nanopore Technologies, Oxford, UK). The library was sequenced on an Mk1C (MinKNOW software version 23.07.12) for 72 h using an R10.4.1 flow cell.

### Tiled ONT sequencing for SARS-CoV-2

A total of 14 SARS-CoV-2 positive extracts, tested at the Robert Koch Institute were selected. These samples originated from IMSSC2-lab networks that were received under the RKI integrated genomic surveillance program. All samples were nasopharyngeal or oropharyngeal swabs originating from patients in Germany from January to February 2024. Total nucleic acid extraction was performed using MagNA Pure 96 DNA and Viral NA Small Volume Kit (Roche Life Science, Mannheim, Germany) on an automated extraction instrument (MagNA Pure 96 system, Roche Diagnostics) according to the manufacturer's manual. Reverse-transcription was performed on viral RNA extracts using LunaScript® RT SuperMix (New England Biolabs, as part of the NEBNext ARTIC SARS-CoV-2 Companion Kit (Oxford Nanopore Technologies) according to the manufacturer's protocol. Amplification was performed with 35 cycles annealing at 60 °C for 2 min and elongation at 72 °C for 3 min for both pools, without amplicon cleanup afterward. Barcoding was performed using the ONT Native Barcoding Expansion kit (EXP-NBD196). Fourteen samples together with 2 negative controls were multiplexed on a FLO-MIN 114 flow cell version R10 and sequenced on a GridION Mk1 device for 16 h.

### RT-qPCR (PV 1–3)

Different varVAMP RT-qPCR assays were performed on serial dilution series of PV 1–3 (Sabin) RNA with RNA concentrations ranging from 1–3 ng/μl of the stock solution to evaluate the performance and sensitivity of the different assays. To prove the lack of cross-detection between the PV vaccine strains, all possible combinations were tested. Quantitative realtime RT-PCR was carried out using the 4X CAPITAL™ 1-Step qRT-PCR Probe Master Mix (BR0502002, Biotechrabbit, Germany) on a Roche instrument, following the manufacturer's instructions. The reaction contained 0.4 μM of each primer with a total reaction volume of 20 μl. qRT-PCR cycling program started with reverse transcription at 50 °C for 10 min followed by an activation step at 95 °C for 3 min and 45 cycles for target amplification (95 °C for 10 s, 59 °C for 30 s). The primer annealing temperature was chosen after preliminary tests with different annealing temperature settings ranging from 56–64 °C on a Biometra TAdvanced (analytik jena, Germany), and the products were observed on an 1.5% agarose gel. The fluorescence was measured during the extension step with three different channel setups with respect to the probe fluorophore (FAM = 465–510 nm, JOE = 533–580 nm, CY5 = 618–660 nm).

### Sequencing data analysis

The de-multiplexed raw Illumina reads were subjected to a custom Galaxy pipeline, which we initially developed for tiled amplicon sequencing of SARS-CoV-2[6]. These reads were preprocessed with fastp (v0.20.1)[67] and mapped to their respective closely related genomes via BWA-MEM[68] (v0.7.17). Importantly, the 3′ and 5′ ends of the viral reference genome were masked prior mapping until the 5′ and 3′ end of the flanking primer binding regions, respectively, as no novel information can be generated in these regions. After mapping, the primer sequences were trimmed with ivar trim (v1.3.1). Variants (SNPs and INDELs) were called with the ultrasensitive variant caller LoFreq[69] (v2.1.5), demanding a minimum base quality of 30 and a coverage of at least 20-fold. Afterwards, the called variants were filtered with a minimum variant frequency of 10% and on strand bias support. Finally, consensus sequences were constructed with BCFtools (v1.15.1)[70]. Regions at both genome ends that lie outside the amplicons, regions with low coverage (<20x) or variant frequencies between 0.3 and 0.7 were masked with Ns.

For Oxford nanopore sequencing of SARS-CoV-2, poreCov (v1.9.3), a Nextflow workflow specifically tailored for SARS-CoV-2 genome reconstruction from nanopore amplicon data, was used to perform mapping (minimap2; v2.17)[71], primer clipping, variant calling (Medaka; v1.8.0), and consensus genome reconstruction[72]. We ran

poreCov to initially filter reads below 400 bp (--minLength 400) and above 1 kbp (--maxLength 1000) while coverage downsampling was disabled (--artic_normalize 0). The r1041_e82_400bps_sup_v4.2.0 model was used for variant calling with Medaka and variant calls were filtered by a minimum base quality of 20 and a coverage of at least 20-fold. We set the allelic frequency of called mutations to 1 to ensure compatibility with the Illumina data for the in silico analysis.

For Oxford nanopore sequencing of HAV samples, the pod5 raw data were duplex basecalled with Dorado version 0.5.3 and demultiplexed. The fastq files were subsequently processed using a custom Galaxy pipeline. First, the raw data were preprocessed with fastp[67] excluding reads <50 bp and >2000 bp. Afterward, the reads were mapped to the HAV reference genome NC_001607 using minimap2 (v2.17)[71] and trimmed with ivar trim (v1.3.1). Variants were called with medaka (v1.3.2) and consensus sequences were constructed with bcftools (v.1.15.1)[70]. Regions at both ends of the genome that lie outside the amplicons and regions with low coverage (<20x) were masked with Ns.

SAMtools v.1.20[70] was used for mapping statistics and to extract unmapped reads from aligned bam files. Next, BEDtools v2.27.1[73] was used to convert bam files to fastq. To identify the origin of unmapped reads, they were classified with Kraken2 v.2.1.3[74] using the Minikraken v2 database.

### Data analysis and visualization
qPCR amplification curves were analyzed using the Roche LightCycler 480 II device software version LCS480 1.5.1.62 (Roche Applied Science). The data were analyzed using the 2nd derivative. Mapped bam files were analyzed and visualized with BAMdash v.0.2.4 (https://github.com/jonas-fuchs/BAMdash). We used GraphPad Prism 8 (genome recovery and qPCR), R 4.3.2 (phylogenetic tree) or python 3.11 (remaining figures) for data analysis and visualization. The data and code to reproduce the figures are available at: https://github.com/jonas-fuchs/varVAMP_in_silico_analysis. The schematic varVAMP workflow and data preparation workflow were created with biorender (https://www.biorender.com).

### Reporting summary
Further information on research design is available in the Nature Portfolio Reporting Summary linked to this article.

## Data availability
Genome recovery raw data (Figs. 2 and 5) and qPCR data are provided with this paper (Source data file). The raw data and code to reproduce the remaining figures in this manuscript is available at https://github.com/jonas-fuchs/varVAMP_in_silico_analysis (https://doi.org/10.5281/zenodo.14826645). All input multiple sequence alignments and varVAMP outputs for primers that have been evaluated in this study are available at: https://github.com/jonas-fuchs/ViralPrimerSchemes (https://doi.org/10.5281/zenodo.10562883). Additionally, primer schemes are available at https://labs.primalscheme.com/. The raw sequencing data have been deposited at ENA under the study accession number: PRJEB74744. Source data are provided with this paper.

## Code availability
varVAMP v.1.2.2 and BAMdash v.0.3.1 are open source and available at https://github.com/jonas-fuchs/varVAMP (https://doi.org/10.5281/zenodo.14826629) and https://github.com/jonas-fuchs/BAMdash (https://doi.org/10.5281/zenodo.13253684). The Galaxy version of varVAMP is available at https://usegalaxy.eu/root?tool_id=toolshed.g2.bx.psu.edu/repos/iuc/varvamp/varvamp/.

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

## Acknowledgements

This project was funded by the Müller-Fahnenberg-Stiftung of the Albert-Ludwigs-Universität Freiburg, Germany, the ZoRaHED project (01KI2103, German Federal Ministry of Education and Research) and the BY-COVID project (101046203, European Union's Horizon Europe research and innovation programme). This project was in part supported by co-funding from the European Union's EU4Health programme under project no. 101113012 (IMS-HERA2). The European Galaxy project is in part supported by the Ministry of Science, Research and the Arts Baden-Württemberg (MWK) within the framework of LIBIS/de.NBI Freiburg. We thank the AMELAG project, financed by the German Federal Ministry of Health, for the financial support of the SARS-CoV-2 primers. We also acknowledge funding of the IMS-RKI project by the German Federal Ministry of Health. We would like to thank Dr. Freya Fleckenstein for many suggestions for the code base of varVAMP and for help with the mathematical part of the "Methods" section. We would also like to thank Dr. Zsolt Ruscics for his valuable and critical input at the early stages of varVAMP development. We thank Ashkan Ghassemi for his help in preparing representative SARS-CoV-2 genomes and Dr. Christian Blumenscheit for help in ordering the SARS-CoV-2 varVAMP primers. We also appreciate the efforts of the submitting laboratories for the procurement and provision of SARS-CoV-2 genomes via the German Electronic Sequence Data Hub and those laboratories providing SARS-CoV-2 positive samples via the IMS-SC2 network. We acknowledge the Sequencing Core Facilities of the Genome Competence Centre at the Robert Koch Institute for providing excellent sequencing services for the PV and SARS-CoV-2 samples, and Aaron Houterman for his excellent technical assistance. Furthermore, we would like to thank Dr. Joshua Quick and Christopher Kent for integrating varVAMP primer schemes in labs.primalscheme.com. We would also like to thank Prof. Dr. Otto Haller for his helpful comments on the manuscript.

## Author contributions

J.F., M.H., M.S., W.M., M.P. conceptualized the project. J.F. wrote the manuscript. J.F. developed varVAMP with help from W.M. W.M., E.B. and B.G. deployed varVAMP to CONDA, DOCKER and Galaxy. T.K. critically evaluated the algorithms employed by varVAMP. M.S. and J.F. selected the input data for primer design. J.F. and J.Kl. designed the primers with varVAMP. J.Kl., M.S., J.Kr., C.W., L.A., L.J., A.M., C.B., M.B., J.P., R.J., J.W., J.S., C.M., S.B. and S.S. generated and evaluated the experimental data. J.F., N.B., J.Kl. and M.H. performed the bioinformatic analyses. J.F. performed the phylogenetic analyses and in silico primer evaluation. All authors reviewed and edited the manuscript.

## Funding

## Competing interests

The authors declare no competing interests.
