## [Transparent Peer Review file · Nature Communications]

varVAMP: degenerate primer design for tiled full genome sequencing and qPCR.

Corresponding Author: Dr Jonas Fuchs

Version 1:

Reviewer comments:

Reviewer #1

(Remarks to the Author)

Review Comments:

The manuscript by Jonas Fuchs et al. presents varVAMP, a tool designed for degenerate primer design with potential applications in tiled full-genome sequencing and qPCR for viral pathogens with high genomic variability. The manuscript is well-written and organized. varVAMP is an innovative tool for designing pan-specific primers that tolerate sequence variation across strains. The utility of varVAMP is demonstrated across various species, and comparisons with PrimalScheme and Olivar are provided. The authors highlight varVAMP's strengths in minimizing primer mismatches and enhancing sequencing accuracy and coverage, especially for highly variable viral genomes.

In conclusion, I believe the paper introduces new features and approaches based on existing tiling primer design algorithms and should be considered for publication in Nature Communications, provided the authors address the comments below. I would be happy to review a revised manuscript.

1. Tiling Primer Design Plexity Upper Limitation:

Considering the clinical application of viral/pathogen detection, the total plexity of tiled primer design for a typical pathogen remains relatively low. What would be the upper limit of plexity that varVAMP can handle? Is there a threshold where the performance of the designed primer sets significantly drops?

For higher plex tiled PCR, primer concentration usually decreases as plexity increases. The authors mention an example of optimizing multiplex PCR primer concentrations but do not go into detail. A more thorough explanation or separate section on how multiplexing can be optimized using varVAMP would be useful, as multiplexing is critical in high-throughput sequencing applications.

2. Methods Comparison:

The manuscript compares varVAMP to PrimalScheme and Olivar, but the comparisons focus mainly on computational performance (e.g., mismatch rates, running times, Supp. Table 1). A more comprehensive comparison, including data from supplementary figures (e.g., alignment coverage, genome recovery), could be summarized in a single table to provide a clearer overall performance evaluation.

3. Handling Hypervariable Regions:

The manuscript mentions that amplicons near hypervariable regions can experience dropouts, which could limit pathogen diagnostics. The authors also show coverage drops in Figure 5 and Supp. Figure 1, with the worst case being a dropout in the Supp. Figure 1 ratHEV panel. Detailed sequence analysis may be needed here to provide a hypothesis for the cause of these dropouts. More discussion on how varVAMP can address or mitigate these challenges would be helpful. Additionally, a re-design algorithm for poorly performing primers based on experimental validation results would improve the manuscript.

4. Uniformity of Tiled PCR Amplicons:

For each tiled PCR amplicon design, the fold difference in reads mapped to each amplicon can vary by more than 2-3 orders of magnitude. Poor uniformity across amplicons leads to a need for higher sequencing depth to ensure adequate genome coverage. From a design perspective, alternative strategies that consider energy penalties at the 3' end could help normalize PCR efficiency. From an experimental standpoint, primer concentration optimization could also improve uniformity

across amplicons.

5. Non-Specific Amplification:

In Figure 2C, the agarose gel analysis results show non-specific amplification products in cluster 4 amplicons 2 and 3. This suggests non-specific amplification during NGS library preparation. The authors should provide overall alignment ratios or mapping ratios for each tiled design NGS run. Additionally, a comprehensive analysis of non-specific amplification (based on unmapped reads) could help optimize the primer design algorithm.

(Remarks on code availability)

I read through most of the files, including the readme, most of Python script, data generation-related files, and primer design output files. I believe the files and readme instructions are clear enough for a researcher in the field to begin implementing the code. Unfortunately, I didn't get a chance to install and run the code.

Reviewer #2

(Remarks to the Author)

This paper presents varVAMP, a novel tool for automated primer design for viral genome sequencing. The approach and results presented here appear a clear advancement over the state-of-the-art for primer design. The analysis presented shows that varVAMP produces primer schemes with fewer mismatches to input genomes, compared to available tools PrimalScheme and Olivar. The results also include extensive testing of the varVAMP primer sequences on real patient samples, showing that the primer schemes are suitable for different use cases.

The manuscript is written very clearly and is practically error-free. Also the software appears to be high-quality and very usable and valuable for the community. It is available through various platforms and easy to install.

My main comments regarding the manuscripts as it stands are regarding the Methods section. While I can understand most of the methodology described under "Software", it was hard to follow without a figure to illustrate the main ideas. Figure 1 shows the main workflow, but it doesn't help in understanding the algorithm design. Several aspects remain unclear:

In line 124, what are the lengths l_1, l_2, \dots, l_m ?

In line 128, "we define that optimization is reached if less than 50% of the alignment can be considered for potential primers". What does this mean, and what is the motivation?

In line 135, about masking gaps: "this ensures that primers will not span regions that could be potential INDEL sites". But N's could also be introduced as ambiguous nucleotides later on, so when in line 156 "the window is closed if ... a gap is reached", how do you actually know? This must depend on the threshold t used, since for high thresholds an N could be introduced in highly variable positions? So is the window closed when any N is reached?

In line 176, "we define position-specific penalties ...". How are these determined, what is the intuition?

In line 248: "the shortest path is first evaluated on the amount of off-target hits (...)". It is not clear to me exactly what this means.

Overall, I think that these parts of the methods section should be refined and would strongly benefit from a (supplementary) figure to illustrate these principles.

Other minor comments:

- Line 43: "studing"  studying
- Line 62: "insertions and deletions sites"  insertion and deletion sites
- Line 579: "... overall higher greater ...". This sentence needs to be fixed?
- The manuscript refers to "kmers" in various places, but commonly this is written as "k-mers".
- Table 1: from the table and caption itself it is not clear what the threshold refers to and how it was determined. It is explained in the text, but would be helpful in the table caption as well.
- Figure 1: first box has user arguments, if I understand correctly "n ambiguous nt" is one argument? If yes, better put it on one line. If not, clarify what n means here.
- In several figures the axis labels and legends are too small to be readable. This holds in particular for figure 1b and all of figure 6.

(Remarks on code availability)

The code is well documented, easy to install, and the github page shows that it is also well supported and used in practice. All data for reproducing results appears to be available.

Reviewer #3

(Remarks to the Author)

[General comments]

The authors describe an open-source tool, varVAMP, for designing tiled amplicons and qPCR assays for viral pathogens. varVAMP takes a multiple sequence alignment (MSA) of a specific virus as input and generates candidate primers/amplicons directly from the consensus, allowing ambiguous bases to be included in the output primers. For tiled amplicons, varVAMP first builds a weighted directed graph of candidate amplicons and searches for the shortest path using Dijkstra's algorithm.

The output primers of varVAMP were tested on viruses with high mutation density, and sequencing results showed that the

viral genomes were successfully recovered. The authors experimentally validated varVAMPS primers for several viruses and patient samples.

By incorporating ambiguous bases and penalizing mismatches, varVAMP outperforms existing tools with respect to highly variable genomes in terms of primer mismatches. However, prioritizing the optimization of mismatches and adding ambiguous bases might cause issues when the number of primers in the same primer pool scales up. PCR byproducts such as primer dimers and non-specific amplifications grow quadratically, leading to a lower read mapping rate. This issue might be exacerbated when sample quality is low and/or has complex background genomes. Nonetheless, varVAMP represents an advance specifically for highly variable viral genomes such as HEV, which are of clinical importance.

[Major comments]

1. The varVAMP design algorithm is not clearly described in the Methods section. Although a primer design algorithm/workflow is often heuristic-based and parameters are typically selected empirically, it is essential to describe all the steps and formulas clearly and explain the reasoning behind each moving part.
2. In line 120, "If n_a and t are not given, t is optimized and $n_a=2$ ", but then it says, " n_a or t are incremented by -1 or 0.1 , respectively", which is a bit confusing to me (not sure whether n_a is optimized or not). The initial value of t is not mentioned either.
3. Line 127: the reasoning behind $l_i + n_a \geq l_{min}$ is unclear. Per the manuscript, l_i is stretch length and n_a is the number of ambiguous nucleotides; is this an ad hoc formula?
4. Why is 50% used for optimization termination? What are the trade-offs of setting this threshold larger or smaller? Is only t changed after each iteration?
5. Line 134: I understand why a different consensus threshold for gaps is needed, but it's not clear to me why it's set to $(1-t)$ instead of a fixed ratio of t (e.g., $0.5*t$). For example, if $t \geq 0.5$, $(1-t) \leq t$; else $(1-t) > t$, which is confusing.
6. Masking gaps with 'N' seems to conflict with nucleotide frequency calculation in the next step since 'N' is also an ambiguous nucleotide (25% each). Also, for MSA columns where gaps are not masked ($\text{gap frequency} \leq (1-t)$), if $(1-t) > t$, what is the consensus nucleotide? e.g., and $\text{gap frequency} = 0.55$.
7. Line 199: The start and stop of an amplicon include both primers, so the overlapping of two amplicons does not guarantee the overlapping of their inserts (sequence between forward and reverse primers). But to cover the whole genome, all inserts should overlap since primers should be removed from sequencing reads during data analysis.
8. Line 209: "the lowest penalized amplicon... is termed v_{max} " is unclear. I assume v_{max} is the last amplicon in the path. Also, the reason for comparing with $stop_{max}$ (line 213) is unclear. Why not just select the path with the highest coverage?
9. Line 216: the definition of heterodimer is not clear. Eliminating complementary regions between primers is impractical when numerous primers are included in the same pool (e.g., 100 primers). Also, is there a termination condition for primer swapping when heterodimers are found?

[Minor comments]

- a) Line 160: I assume x_{1+1} should be x_{m+1} , where $m=n_a$?
- b) Line 171: are self-heterodimers considered here?
- c) Line 175: It is rather unclear what "permutation penalty" means here. pp is defined as permutation penalty: "permutation penalty is calculated by multiplying permutation penalty".
- d) Line 203: generated -> generates
- e) Line 208: $v_i \in G \rightarrow v_i \in V$
- f) Line 545: Fig. 2a -> Fig. 2d?
- g) Line 579: "...with overall higher greater in which..." typo?

(Remarks on code availability)

I inspected the code and command-line tool. varVAMP is open source and well-documented on GitHub. It can be installed locally or accessed online via Galaxy. I was able to install it via Conda and run the test MSA successfully.

Version 2:

Reviewer comments:

Reviewer #1

(Remarks to the Author)

I thank the authors for the updates and additions to the manuscript. I believe most of the key concerns have been addressed through the revisions and the new language in the text. I am satisfied with the data and the scientific components presented. Therefore, I would like to recommend the manuscript for publication in Nature Communications.

(Remarks on code availability)

Point-by-point response

REVIEWER COMMENTS

Reviewer #1 (Remarks to the Author):

Review Comments:

The manuscript by Jonas Fuchs et al. presents varVAMP, a tool designed for degenerate primer design with potential applications in tiled full-genome sequencing and qPCR for viral pathogens with high genomic variability. The manuscript is well-written and organized. varVAMP is an innovative tool for designing pan-specific primers that tolerate sequence variation across strains. The utility of varVAMP is demonstrated across various species, and comparisons with PrimalScheme and Olivar are provided. The authors highlight varVAMP's strengths in minimizing primer mismatches and enhancing sequencing accuracy and coverage, especially for highly variable viral genomes.

In conclusion, I believe the paper introduces new features and approaches based on existing tiling primer design algorithms and should be considered for publication in Nature Communications, provided the authors address the comments below. I would be happy to review a revised manuscript.

We would like to thank the reviewer for his thorough and comprehensive revision of our manuscript. Please refer in the following to our point-to-point response. Following your comments, we have marked changes in the manuscript in yellow.

1. Tiling Primer Design Plexity Upper Limitation:

Considering the clinical application of viral/pathogen detection, the total plexity of tiled primer design for a typical pathogen remains relatively low. What would be the upper limit of plexity that varVAMP can handle?

1. In principle, there is no computational limit to the plexity. However, highly conserved alignments will take more time to compute as the resulting amplicon graphs (Supplementary Figure 1) get very large and complex. Dijkstra's search is then the main component that defines the time complexity of varVAMP. To exemplify this, we ran varVAMP on an Mpox alignment with minimal differences (see Reviewer Figure 1) resulting in a large graph with around 120k amplicons as nodes, which required a high computation time of roughly 10 h. varVAMP was developed for creating pan-specific primer schemes on alignments with high variability. Higher variability inherently restricts the number of found primers and therefore requires less computation time due to a lower graph size. Therefore, computation time is not directly linked to plexity but to alignment variability. Importantly, we would like to acknowledge that other primer design software, such as PrimalScheme, is more suited for primer design on conserved alignments, which we also emphasize in the Discussion in the manuscript.

Higher plexity also results in higher chance of heterodimer formation within pools. Although this is something we would like to rework in future versions, varVAMP only attempts one initial heterodimer solve between primers of different amplicons within one pool by switching to primers that have been excluded in the 'select best primers' step (see Methods

and Supplementary Figure 1). Please also refer to Reviewer 3, Point 30 for a more extensive explanation.

Heterodimer considerations and computation time for highly conserved and/or large alignments are varVAMP's main trade-offs for allowing its pan-specificity design approach. We now discuss the software's limitation in more detail in line 830-852.

Reviewer Figure 1. varVAMP design on a Mpox alignment with minimal variability. 1.4 million k-mers were tested for their primer suitability resulting in a final set of ~10k primers and ~120 k possible amplicons. The final design consisted of 182 amplicons and took roughly 10 h to compute.

Is there a threshold where the performance of the designed primer sets significantly drops?

2. It is extremely difficult to give a definite answer to this question, as this is a multifactorial problem and not easy to test. Importantly, performance is not necessarily restricted to the threshold parameter. It also depends on the $-a$ parameter, as a higher number of ambiguous characters results in a higher permutation per primer which then might affect PCR efficacy (e.g. due to a higher number of potential off-target amplifications). Moreover, it can also depend on the sample the primer scheme is tested on (e.g. how many mismatches for each primer are present in the target sequence or how high the template load is). Solely based on our experience, we would rather associate varVAMP's performance to the mean pairwise sequence identity of an alignment (Table 1). So far, we had decent results with a mean pairwise sequence identity over 70 %. This is why we showed our ratHEV design within the paper. There are good indicators that in such a case (57 % pairwise sequence identity), varVAMP will design poorly performing primers because:

- i) primers were highly permuted (Figure 3 b)
- ii) primers mismatched the most with the input alignment (Figure 4 a)
- iii) the scheme had the worst performance with tested samples (Figure 6 a/b and Suppl. Figure 2)
- iv) the scheme had the highest amount of off-targets (Suppl. Figure 5 a)

We acknowledge these points in the results and discussion. However, we do not think that we have enough empirical evidence to define a precise cut-off.

For higher plex tiled PCR, primer concentration usually decreases as plexity increases. The authors mention an example of optimizing multiplex PCR primer concentrations but do not go into detail. A

more thorough explanation or separate section on how multiplexing can be optimized using varVAMP would be useful, as multiplexing is critical in high-throughput sequencing applications.

3. This is closely linked to Point 6 (uniformity of tiled amplicons). We apologize, as this was not clearly written in the text. varVAMP cannot be used to computationally optimize PCR primer concentrations. Our primer optimizations in the case of Polio were manual work. Based on the relative coverages over the reference, we adjusted the primer concentrations to each other and again evaluated the uniformity by sequencing. For Polio, we did two rounds of primer adjustment (primer adjustments and resulting coverages are depicted in Suppl. Figure 4). We now clarified this in lines 696-701.

2. Methods Comparison:

The manuscript compares varVAMP to PrimalScheme and Olivar, but the comparisons focus mainly on computational performance (e.g., mismatch rates, running times, Supp. Table 1). A more comprehensive comparison, including data from supplementary figures (e.g., alignment coverage, genome recovery), could be summarized in a single table to provide a clearer overall performance evaluation.

4. We now provide a more extensive table (Table 3) summarizing all findings of the comparison and scraped supplementary table 1 to avoid duplication. We refer to it in the text accordingly (lines 628, 632 and 641).

3. Handling Hypervariable Regions:

The manuscript mentions that amplicons near hypervariable regions can experience dropouts, which could limit pathogen diagnostics. The authors also show coverage drops in Figure 5 and Supp. Figure 1, with the worst case being a dropout in the Supp. Figure 1 ratHEV panel. Detailed sequence analysis may be needed here to provide a hypothesis for the cause of these dropouts. More discussion on how varVAMP can address or mitigate these challenges would be helpful. Additionally, a re-design algorithm for poorly performing primers based on experimental validation results would improve the manuscript.

5. The reviewer raises a valuable point and we now did a detailed sequence analysis. Notably, our statement to the hypervariable region was to indicate that in this region a high sequence plasticity and variability is commonly observed and might have caused the dropouts (we rephrased this in lines 579-580). Previously, we showed that most primer sequences have no or only one mismatch with the target sequence with a few isolated cases where we saw two mismatches (Figure 6 b). Interestingly, for all dropouts one primer showed two mismatches with the target sequence. In all cases, one of those mismatches was also located within the last five bases of the 3' end potentially disturbing primer binding stability. Moreover, we analysed the respective positions in the input alignment and show that these variations are found within the alignment but not in a high enough frequency to get integrated into the primer sequence. This analysis is now shown in Supplementary Figure 5 (described in lines 706-712). Based on the SNP frequency in our alignment, our target sequence had a

combination of two rare mutations. We hypothesize that these mismatches caused the dropouts (we now discuss this in lines 802-807). Although, this is now very speculative it might very well be that our input data selection does not reflect the actual stoichiometry of the HEV variability and that these variations are in reality more frequent. On the other hand, it might also be that we were indeed just unlucky and our test samples had rare genomic variations.

Therefore, we fully agree a re-design algorithm would help in such cases and this is something that we would like to implement in the future. We are thinking about graph node switching which could allow switching to excluded amplicons while staying within the graph. However, this enhancement is beyond the scope of this revision as it could take a lot of new code and/or code refactoring. Currently, it is only possible to manually switch to other primers. varVAMP reports all potential primers irrespective if they are selected for the final scheme.

4. Uniformity of Tiled PCR Amplicons:

For each tiled PCR amplicon design, the fold difference in reads mapped to each amplicon can vary by more than 2-3 orders of magnitude. Poor uniformity across amplicons leads to a need for higher sequencing depth to ensure adequate genome coverage. From a design perspective, alternative strategies that consider energy penalties at the 3' end could help normalize PCR efficiency. From an experimental standpoint, primer concentration optimization could also improve uniformity across amplicons.

6. We agree that energy considerations at the three prime end are important. varVAMP indirectly considers energy at the 3' end by a user definable GC content for the last 5 primer bases (commonly known as GC clamp – see line 182). This is now also visualized in Suppl. Figure 1. Moreover, we also try to minimize mismatches in these last 5 bases (see lines 182-191) to strengthen primer binding.

Poor uniformity across amplicons is a common problem in all amplicon schemes and to our knowledge it is not well understood what the main drivers are and how to predict PCR efficacy *in silico*. To compensate for inefficient amplicon performance, researchers commonly switch primers or indeed adjust primer concentrations. Our goal was to show, how varVAMP primers would perform without any additional experimental adjustment. Nevertheless, we also show that primer concentrations can be manually optimized leading to better uniformity in the case of our Polio1-3 scheme (Suppl. Figure 4). Please also refer to Point 3.

5. Non-Specific Amplification:

In Figure 2C, the agarose gel analysis results show non-specific amplification products in cluster 4 amplicons 2 and 3. This suggests non-specific amplification during NGS library preparation. The authors should provide overall alignment ratios or mapping ratios for each tiled design NGS run. Additionally, a comprehensive analysis of non-specific amplification (based on unmapped reads) could help optimize the primer design algorithm.

7. We would like to thank the reviewer for this question, as we have not evaluated this systematically. We know re-ran our analyses and extracted the mapping information with samtools stats and evaluated the off-targets in unmapped reads with kraken2 (method described in line 504-507). For the large majority of samples, we show that over 80 % of reads mapped to the reference. The off-target reads were mostly of human or bacterial origin. Differences within schemes are largely dependent on the sample type as clinical samples showed higher amounts of non-mapped reads. We show the results of this analysis in figure 6 d, supplementary figure 4 and describe the results lines 720-727, 808.

Reviewer #1 (Remarks on code availability):

I read through most of the files, including the readme, most of Python script, data generation-related files, and primer design output files. I believe the files and readme instructions are clear enough for a researcher in the field to begin implementing the code. Unfortunately, I didn't get a chance to install and run the code.

Reviewer #2 (Remarks to the Author):

This paper presents varVAMP, a novel tool for automated primer design for viral genome sequencing. The approach and results presented here appear a clear advancement over the state-of-the-art for primer design. The analysis presented shows that varVAMP produces primer schemes with fewer mismatches to input genomes, compared to available tools PrimalScheme and Olivar. The results also include extensive testing of the varVAMP primer sequences on real patient samples, showing that the primer schemes are suitable for different use cases.

The manuscript is written very clearly and is practically error-free. Also the software appears to be high-quality and very usable and valuable for the community. It is available through various platforms and easy to install.

We would like to thank the reviewer for his thorough and comprehensive revision of our manuscript, particular the method section. Your constructive criticism overlapped in many cases with Reviewer 3. We tried to merge answers to avoid redundancy and refer to the respective points where we provide a detailed answer. Based on your comments, we have marked changes in the manuscript in blue.

My main comments regarding the manuscripts as it stands are regarding the Methods section. While I can understand most of the methodology described under "Software", it was hard to follow without a figure to illustrate the main ideas. Figure 1 shows the main workflow, but it doesn't help in understanding the algorithm design.

8. Please refer to Point 14 and Reviewer 3, Point 22.

Several aspects remain unclear:

In line 124, what are the lengths l_1, l_2, \dots, l_m ?

9. Please refer to Reviewer 3, Point 24.

In line 128, "we define that optimization is reached if less than 50% of the alignment can be considered for potential primers". What does this mean, and what is the motivation?

10. Please refer to Reviewer 3, Point 25.

In line 135, about masking gaps: "this ensures that primers will not span regions that could be potential INDEL sites". But N's could also be introduced as ambiguous nucleotides later on, so when in line 156 "the window is closed if ... a gap is reached", how do you actually know? This must depend on the threshold t used, since for high thresholds an N could be introduced in highly variable positions? So is the window closed when any N is reached?

11. The reviewer is of course correct; this would not work as described. We forgot to describe, that in the alignment pre-processing all characters are converted to lower case prior gap cleaning. Common gaps on the other hand are marked in a capital 'N'. The consensus sequences inherit the letter case and our region search algorithm is case sensitive and can distinguish between 'n' (atcg) and 'N' (small gaps) or 'NN' (large gap). We now state this in line 137-138 and 163-164. Moreover, we visualize this idea in Supplementary Figure 1.

In line 176, "we define position-specific penalties ...". How are these determined, what is the intuition?

12. The intuition is that the last five bases of the primer receive penalty multipliers that are used to multiply the frequency of mismatches at that position. Moreover, we set the highest standard mismatch multiplier for the very 3' end position, the second highest for the 3' end -1 position and so on in order to minimize mismatches at the 3' end (Figure 4b). Importantly all penalty multipliers are user definable within varVAMP. We now state this in line 190-191. For the standard settings, we tried to balance the penalties for the 3' mismatch penalty to the base penalty and permutation penalty so that none of the penalties is over-represented in the final primer penalty.

In line 248: "the shortest path is first evaluated on the amount of off-target hits (...)". It is not clear to me exactly what this means.

13. Our edge weights in the graph consists of a tuple (off-target true/false 1/0, amplicon penalty). When considering the shortest path for a current node, we asked whether the current distance (cumulative off-targets, cumulative penalty) is less then a previously considered path that also leads to this node. Here, we make use of how python compares tuples as $(2, 2) < (3, 1)$ is True but $(2, 2) < (1, 3)$ is False. The first index (off-targets) is compared before the second (amplicon penalty). For a code reference of our Dijkstra implementation, please refer to scripts/scheme.py line 136-164. We shortly explained this in the prior section for the graph generation. However, we clarified this now additionally in lines 261-263 and also visualize this idea with an example in Supplementary Figure 1.

Overall, I think that these parts of the methods section should be refined and would strongly benefit from a (supplementary) figure to illustrate these principles.

14. We agree and based on your and Reviewer 3 comments, we have reworked parts of the method section and provide an extensive supplementary figure that visually describes the main steps of the tiled scheme generation (Supplementary Figure 1). (Please also refer to Reviewer 3, Point 22)

Other minor comments:

- Line 43: "studing"  studying

15. Fixed in line 43.

- Line 62: "insertions and deletions sites"  insertion and deletion sites

16. Fixed in line 62.

- Line 579: "... overall higher greater ...". This sentence needs to be fixed?

17. Yes thank you! Fixed in line 598.

- The manuscript refers to "kmers" in various places, but commonly this is written as "k-mers".

18. Fixed in lines 111, 172, 174, 178, 236, 237, 527, 532.

- Table 1: from the table and caption itself it is not clear what the threshold refers to and how it was determined. It is explained in the text, but would be helpful in the table caption as well.

19. Fixed in the table caption (lines 1089-1091).

- Figure 1: first box has user arguments, if I understand correctly "n ambiguous nt" is one argument? If yes, better put it on one line. If not, clarify what n means here.

20. Correct this is the -a argument. Fixed for Figure 1.

- In several figures the axis labels and legends are too small to be readable. This holds in particular for figure 1b and all of figure 6.

21. We agree with the reviewer's assessment and reworked the figure text sizes in 1b and figure 6. We have additionally increased it in Figure 4.

Reviewer #2 (Remarks on code availability):

The code is well documented, easy to install, and the github page shows that it is also well supported and used in practice. All data for reproducing results appears to be available.

Reviewer #3 (Remarks to the Author):

[General comments]

The authors describe an open-source tool, varVAMP, for designing tiled amplicons and qPCR assays for viral pathogens. varVAMP takes a multiple sequence alignment (MSA) of a specific virus as input and generates candidate primers/amplicons directly from the consensus, allowing ambiguous bases to be included in the output primers. For tiled amplicons, varVAMP first builds a weighted directed graph of candidate amplicons and searches for the shortest path using Dijkstra's algorithm.

The output primers of varVAMP were tested on viruses with high mutation density, and sequencing results showed that the viral genomes were successfully recovered. The authors experimentally validated varVAMP's primers for several viruses and patient samples.

By incorporating ambiguous bases and penalizing mismatches, varVAMP outperforms existing tools with respect to highly variable genomes in terms of primer mismatches. However, prioritizing the optimization of mismatches and adding ambiguous bases might cause issues when the number of primers in the same primer pool scales up. PCR byproducts such as primer dimers and non-specific amplifications grow quadratically, leading to a lower read mapping rate. This issue might be exacerbated when sample quality is low and/or has complex background genomes. Nonetheless, varVAMP represents an advance specifically for highly variable viral genomes such as HEV, which are of clinical importance.

We would like to thank the reviewer for his thorough and comprehensive revision of our manuscript, particular the method section. We completely agree with the reviewers assessment that our mismatch optimization can lead to other issues. Our overall assumption was that reduction of mismatches has to be prioritized in the case of high variability to ensure primer binding before minimizing other important factors such as primer dimers. Based on your comments, we discuss these shortcomings in more detail and marked changes in the manuscript in red. To reduce redundancy in our response, we sometime refer to answers of Reviewer 2 or vice versa if points are related.

[Major comments]

1. The varVAMP design algorithm is not clearly described in the Methods section. Although a primer design algorithm/workflow is often heuristic-based and parameters are typically selected empirically, it is essential to describe all the steps and formulas clearly and explain the reasoning behind each moving part.

22. Based on the reviewer responses (especially Reviewer 2 and 3), we reworked parts of the method section. Moreover, we now provide an extensive supplementary figure (Supplementary Figure 1) that visually describes the main steps of the tiled amplicon scheme generation.

2. In line 120, "If n_a and t are not given, t is optimized and $n_a=2$ ", but then it says, " n_a or t are incremented by -1 or 0.1 , respectively", which is a bit confusing to me (not sure whether n_a is optimized or not). The initial value of t is not mentioned either.

23. We apologize, this was not clearly written. It is possible to provide a fixed value for $-a$ or $-t$ or no fixed value for both parameters. In the last case, varVAMP assumes a default fixed value for $-a$ (2). If one value is fixed, then the other value is estimated. For $-a$ we start at the maximum primer length (default = 24, all characters of a primer are permuted) and

increment by -1 and for $-t$ we start at 0.1 and increase it by 0.01. We now clarify this in lines 119-124.

3. Line 127: the reasoning behind $l_i + n_a \geq l_{min}$ is unclear. Per the manuscript, l_i is stretch length and n_a is the number of ambiguous nucleotides; is this an ad hoc formula?

24. Both Reviewer 2 and 3 have found a mistake in our methods section as the given formula does not describe the algorithm correctly. Thank you for pointing this out! We have adjusted this in line 125-127.

Furthermore, we would like to give a more detailed explanation of this algorithm. To estimate the alignment coverage solely based on the input alignment without any pre-processing, we first calculate all nucleotide frequencies at a given position. Next, we extract the highest frequency per position, leaving us with a one-dimensional array in the length of the alignment.

Next, we count how many sequential positions are above the threshold until the first position that falls below the threshold. This resulting stretch length is l_i . For each stretch length, we now add the next $n+1$ stretch lengths, with n being the number of ambiguous characters that we tolerate within a primer. If the sum of those stretch lengths is above the min length of a primer, we retain the current stretch. The coverage estimate is then the sum of retained stretches divided by the alignment length.

Importantly, we call this coverage estimate as it likely differs from the primer region search (Supplementary Figure 1) because we calculate this on the gapped alignment, which might be significantly larger than the pre-processed alignment.

4. Why is 50% used for optimization termination? What are the trade-offs of setting this threshold larger or smaller? Is not t changed after each iteration?

25. We would like to thank both Reviewer 2 and 3 for this question, as the reasoning for the 50 % termination was not explained within the manuscript.

Estimating the $-t$ or $-a$ parameters a priori is rather challenging, as different alignment specific factors directly influence the number of potential primers (variability, GC content), the number of amplicons (target size and overlap) or the distribution of amplicons (e.g. regions with vastly different entropies within the same alignment). Nevertheless, we wanted to provide the user with an option that would give a good estimate solely based on the input alignment. Therefore, we roughly calculate how much of the alignment can be considered for primers. With some manual parameter testing, we found that the estimates for the $-t$ or $-a$ parameters are close to manually optimized parameters if 50 % of the alignment can be considered for potential primers (we know state this within the manuscript in lines 129-131). To challenge the initial observation that was largely based on our test alignment, we now performed automated parameter testing. Therefore, we iteratively adjusted the coverage cut-off between 0.1 and 0.95 in 0.05 increments within the code basis (scripts/param_estimation.py; line 80 and line 90) and re-ran varVAMP for each alignment and coverage cut-off with the same settings as in the paper with the exception of the $-t$ parameter that was estimated by varVAMP (in total 18 re-runs per alignments). The results are shown in Reviewer Figure 2. This analyses demonstrates that a low coverage cut-off results in a high estimate for the threshold. A high threshold results in a low coverage as only a few primers are found. By increasing the coverage cut-off, the threshold is gradually

decreased while the amplicon coverage over the alignment increases. The ‘sweetspot’ is the highest threshold at which the maximal coverage over the alignment is achieved. This is also what we aim for when doing manual $-t$ or $-a$ parameter optimization for a new primer design. For most of the alignments, our initial observations holds true as a cut-off value of 50 % estimates the $-t$ parameter close to the manually optimized parameters (see Table 1). For a code reference, please refer to scripts/param_estimation.py line 53 and following. The criticism of both Reviewer 2 and 3 is highly valuable. Although, the parameter estimation module is not an essential gear of varVAMP (now clarified in line 109), it was written to make the software more user friendly. Importantly, we would like to acknowledge that it is likely that this module will not give an correct estimate for all possible input alignments and user settings. We always recommend manual parameter optimization when running varVAMP as stated in line 770 and lines 135-136. Some of our future ideas for varVAMP include to re-work the parameter estimation module and increase its parameter estimation accuracy.

Reviewer Figure 2. Automated parameter testing by iteratively changing the cut-off for optimization termination and estimating the $-t$ parameter. The main readout was the achieved alignment coverage (amplicons covering the alignment) at the estimated threshold.

No, not only $-t$ is changed. Based on our explanation in Point 24, we hope that this clarifies that either the $-a$ or $-t$ parameter can be changed iteratively and based on the iteratively changed parameter and the fixed parameter the coverage is estimated.

5. Line 134: I understand why a different consensus threshold for gaps is needed, but it's not clear to me why it's set to $(1-t)$ instead of a fixed ratio of t (e.g., $0.5*t$). For example, if $t \geq 0.5$, $(1-t) \leq t$; else $(1-t) > t$, which is confusing.

26. We would like to clarify this in more detail. The consensus threshold describes at a certain alignment position how many sequences (threshold*number of sequences) must have any nucleotide character to reach the threshold. Individual nucleotide frequencies are added from a sorted list until the threshold is reached. If the final list of added nucleotides has more than one nucleotide in it, the ambiguous nucleotide character is chosen in the consensus sequence. $(1-t)*\text{number of sequences}$ in line 134 is the inverse threshold for gaps. This is necessary because if there are more sequences than $(1-t)*\text{number of sequences}$ that have a gap there is not enough nucleotide information at the respective position to reach the desired t .

As gaps do not necessarily have common start and stop positions in an alignment, we perform gap cleaning beforehand by calculating overlapping gap regions in all sequences and then mask stretches that have a higher frequency than $(1-t)*\text{number of alignment sequences}$. For the consensus creation, we then differentiate between masked gaps ('N'/'NN') and non-masked gaps ('-').

To clarify this in the manuscript, we have added a more detailed explanation in lines 139-140 as well as visualized in Supplementary Figure 1.

6. Masking gaps with 'N' seems to conflict with nucleotide frequency calculation in the next step since 'N' is also an ambiguous nucleotide (25% each). Also, for MSA columns where gaps are not masked (gap frequency $\leq (1-t)$), if $(1-t) > t$, what is the consensus nucleotide? e.g., and gap frequency=0.55.

27. Please refer for the first question to Reviewer 2, Point 11. For the second question please also refer to the prior answer (Point 26). For the consensus nucleotide the frequency of a non-masked gap is not included as there is enough nucleotide information to reach the threshold either with a single nucleotide or a combination of variants found at this position. We also visualized this case in Supplementary Figure 1.

7. Line 199: The start and stop of an amplicon include both primers, so the overlapping of two amplicons does not guarantee the overlapping of their inserts (sequence between forward and reverse primers). But to cover the whole genome, all inserts should overlap since primers should be removed from sequencing reads during data analysis.

28. The reviewer raises a valuable point. We only consider overlaps between amplicons and issue a warning if the target overlap is below 50 which is $\sim 2 * \text{max primer size}$. As primer trimming is common practise and necessary, we now enforce insert overlap with varVAMP 1.2.2. We state this in the methods (lines 217-219) and depict the new overlap rationale between amplicons in Supplementary Figure 1. Thank you for this enhancement.

8. Line 209: "the lowest penalized amplicon...is termed v_{max} " is unclear. I assume v_{max} is the last amplicon in the path. Also, the reason for comparing with stop_{max} (line 213) is unclear. Why not just select the path with the highest coverage?

29. Vmax is indeed the last amplicon in the path. However, our Dijkstra search implementation is blind to the location of the amplicons. There can be multiple potential target nodes that e.g. have the same reverse primer and can be reached given the current start node. Therefore, our target node vmax is the lowest penalized of potential target nodes with the most right position in the alignment. With vmax as the target node, we ensure that the alignment coverage is maximized while the amplicon penalties (cumulative edge weights) are minimized when backtracking the shortest path.

The stopmax definition is equally important, as it is likely that varVAMP has to re-run the Dijkstra search multiple times (now explicitly stated in line 225). As varVAMP aims to maximize alignment coverage, our first Dijkstra search starts with the lowest penalized amplicon that has the most left primer. However, this does not guarantee that the resulting path is already optimal (meaning lowest penalized with highest coverage). If in the first search we reach a node with stopmax, there is no need to check if there are better paths given other start nodes. However, if the search only reaches an alignment coverage of 20 % we have to check if we can achieve a higher coverage given another start node. So Dijkstra search is re-run with the start node that has the next most left primer. The comparison with stopmax is for search termination as it is computationally costly to check the shortest path for all nodes.

To clarify this, we illustrated the graph search in Supplementary Figure 1.

9. Line 216: the definition of heterodimer is not clear. Eliminating complementary regions between primers is impractical when numerous primers are included in the same pool (e.g., 100 primers). Also, is there a termination condition for primer swapping when heterodimers are found?

30. The reviewer raises a valuable point. Primer swapping is impractical and primer dimer considerations are in our opinion the main shortcoming of varVAMP.

Firstly, varVAMP only does one initial solve for heterodimers (now stated in 232-233) and accepts if heterodimers are unsolvable because there are no alternatives or new heterodimers occur during primer switching. This is indeed a larger problem for higher plexity schemes.

Secondly, the definition for homodimer/heterodimer is solely based on the thermo-calculations performed with primer3 and the respective user-definable melting temperature cut-off above which primer dimers (either homo- or hetero-dimers) are not considered (stated now in line 176-178). We are aware that this solution is not as sophisticated as other primer design software packages or algorithms (we now acknowledge that this is an additional benefit of Olivar in line 796-797). The main reason for this is that with higher alignment variability, the amount of primers that can be considered for the final design gets less. More stringent primer dimer considerations will require a lower threshold to find enough compatible primers. This will result in more potential mismatches. As our main goal was to minimize primer mismatches, we think that varVAMP's shortcomings for primer dimer considerations are acceptable particular for lower plexity primer schemes. Notably, dimer thermo-calculations with primer3 are structure based. So by decreasing the primer dimer temperature cut-off in the standard configuration (scripts/default_config.py), primer dimers are more stringently handled.

Primer dimers are differently handled for a qPCR design. qPCRs are highly sensitive to primer dimers and can lead to false positive signals particular if probes and flanking primers

hybridize. Here, we additionally consider nucleotide overlaps between the ends of a probe and all permutations of its flanking primers (stated in lines 249-249). Future goals for varVAMP included to address how primer dimers are handled. We now discuss varVAMPs shortcomings including Point 1 of Reviewer 1 in detail in lines 830-852.

[Minor comments]

a) Line 160: I assume x_{1+1} should be x_{m+1} , where $m=na$?

31. This is a good point and actually describes our initial implementation. However, x_{1+1} correctly describes our current implementation. We fully intend to generate overlapping windows as non-overlapping windows miss quite a few potential regions. Please also refer to Suppl. Figure 1 where we visualized such a case. To avoid duplicated k-mers, we later on only consider k-mers with unique consensus coordinates.

b) Line 171: are self-heterodimers considered here?

32. Homodimers are considered already prior to this (line 176). We hard-filter all self-dimers in the initial primer evaluation.

c) Line 175: It is rather unclear what “permutation penalty” means here. pp is defined as permutation penalty: “permutation penalty is calculated by multiplying permutation penalty”.

33. We apologize for the confusing statement. We clarified this in line 186. The permutation penalty is the number of permutations * permutation penalty multiplier.

d) Line 203: generated -> generates

34. Fixed in line 215.

e) Line 208: $v_i \in G$ -> $v_i \in V$

35. Fixed in line 221.

f) Line 545: Fig. 2a -> Fig. 2d?

36. The 2a reference is for the methodology reference (separate alignment of clusters with MAFFT). We now clarified in line 564 that this is not a PCR reference.

g) Line 579: “...with overall higher greater in which...” typo?

37. Please refer to Point 17 of Reviewer 2.

Reviewer #3 (Remarks on code availability):

I inspected the code and command-line tool. varVAMP is open source and well-documented on GitHub. It can be installed locally or accessed online via Galaxy. I was able to install it via Conda and run the test MSA successfully.